# A multi-tiered mechanical mechanism shapes the early neural plate

Angus Inman[1], Elisabeth Spiritosanto [1], Bridget L. Evans [1], Judith E. Lutton[2], Masazumi Tada [3], Till Bretschneider [2], Pierre A. Haas [4,5,6] ✉ & Michael Smutny [1] ✉

The formation of complex tissues during embryonic development requires an intricate spatiotemporal coordination of local mechanical processes regulating global tissue morphogenesis. Here, we uncover a novel mechanism that mechanically regulates the shape of the anterior neural plate (ANP), a vital forebrain precursor, during zebrafish gastrulation. Combining in vivo and in silico approaches we reveal that the ANP is shaped by global tissue flows regulated by distinct force-generating processes. We show that mesendoderm migration and E-cadherin-dependent differential tissue interactions control distinct flow regimes in the neuroectoderm. Initial opposing flows lead to neuroectoderm cell internalisation and progressive multilayer tissue folding which in turn provide forces driving ANP tissue reshaping. We find that convergent extension is dispensable for internalisation but required for ANP tissue extension. Our results highlight how spatiotemporal regulation and coupling of different mechanical processes between tissues in the embryo control the first internalisation and folding events of the developing brain.

Correct shaping and patterning of tissues and organs is a fundamental process during embryonic development which is driven by coordinated cell and tissue level events such as cellular rearrangements and acquisition of specialised cell fates. While biochemical signalling and gene regulation control cell fate specification and tissue patterning, physical forces also play a key role as direct drivers of tissue shape[1–3]. Notably, mechanical forces regulate local and global morphogenetic events during embryonic development, including tissue movement, shape changes, growth and folding[4–6]. Understanding the formation of dynamic three-dimensional (3D) tissues remains especially challenging due to the coordination of sequential cellular rearrangements along multiple body axes within the tissue and interactions with cells from neighbouring tissues[7–10]. Robust tissue morphogenesis requires orchestrated spatiotemporal control of both intrinsic stresses, such as generated by the cellular actomyosin network and extrinsic stresses, including those produced by neighbouring cells or fluid forces[6,11]. In

particular, coupling of forces between cells within a tissue or between adjacent tissues is crucial to drive large-scale tissue remodelling events, including cell intercalations during convergent extension, tissue-wide cellular flows and tissue folding events[12–18]. Notably, these large-scale rearrangements do not function in isolation but can generate forces that feedback into tissue morphogenesis and patterning. For example, a force gradient established in response to FGF8 morphogen signalling in the hindgut-forming endoderm has been shown to drive collective movements of non-contractile cells in the chick embryo[19].

The brain is one of the most complex tissues and undergoes continuous shape changes from the earliest stages of development. In particular, the vertebrate neural plate, a vital precursor of the central nervous system, undergoes extensive rearrangements during neurulation to form the neural tube and nerve cord[20–22]. The morphogenetic events during this process have mostly been studied in amniotes

[1]Centre for Mechanochemical Cell Biology and Division of Biomedical Sciences, Warwick Medical School, University of Warwick, Coventry, UK. [2]Department of Computer Science, University of Warwick, Coventry, UK. [3]Department of Cell and Developmental Biology, University College London, London, UK. [4]Max Planck Institute for the Physics of Complex Systems, Dresden, Germany. [5]Max Planck Institute of Molecular Cell Biology and Genetics, Dresden, Germany. [6]Center for Systems Biology Dresden, Dresden, Germany. ✉e-mail: haas@mpi-cbg.de; michael.smutny@warwick.ac.uk

(mouse and chick) and amphibians and are highly conserved. Convergence movements narrow and elongate the neural plate, followed by bending of the neural plate achieved through internal hinge-points and the adjacent non-neural ectoderm that together support fusion of the neural folds to form the neural tube[22]. This contrasts with teleost neurulation in zebrafish embryos, where the neural plate converges and internalises to form a solid rod that gives raise to the neural tube by cavitation[23]. Despite extensive research into the mechanisms underlying neural tube formation during neurulation, little is currently known about neural plate morphogenesis during gastrulation[20,24,25]. However, failures of functional neural plate formation in the gastrula result in aberrant tissue shape and position and are associated with severe defects of the brain and nervous system in later stages, as evidenced by many mutants identified in *Danio rerio* (zebrafish)[26–28]. Major efforts have been directed at constructing neural plate fate maps[29,30] and to dissect the various morphogen signalling pathways and downstream transcription factors regulating neural plate regionalisation[31]. Yet, our understanding about the underlying mechanisms driving cell and tissue morphogenesis during early neural plate formation in the gastrula remains very limited.

Here, we address this question by investigating how the anterior neural plate (ANP), a precursor of the forebrain, is shaped during zebrafish gastrulation. Using a combination of experiments and simulations, we show that ANP cell and tissue morphogenesis is regulated by a spatiotemporally controlled sequence of extrinsic forces and mechanical tissue coupling and reveal the underlying mechanisms controlling the earliest event of tissue folding in the future forebrain. Our findings highlight the importance of mechanical coordination of distinct morphogenetic events beyond tissue boundaries as a major regulator of shaping complex multi-layered tissues in the developing embryo.

## Results

### The early ANP is dramatically reshaped during late gastrulation

Previous studies suggest that neuroectoderm progenitors show limited cellular rearrangements after regional specification during the early formation of the anterior neural plate (ANP)[24,29]. However, ANP tissue shape is not preserved during gastrulation[32], which raises the intriguing question of how cell and tissue dynamics are coupled to shape the ANP. To address this problem, we studied ANP tissue dynamics throughout gastrulation by using *Tg(otx2:Venus)* transgenic zebrafish embryos[33] that allow visualisation of tissue-specific neuroectoderm progenitor cell fate specification in the prospective ANP (Fig. 1a, b). We observed otx2:Venus expression from ~7.5 h post fertilisation (hpf) onward, with neuroectoderm progenitor cells initially displaying a scattered distribution and irregular tissue outlines (Supplementary Movie 1, Fig. 1b). Analysis of the overall tissue dynamics revealed that little tissue shape changes occurred prior to 8.5 hpf (Fig. 1c–e). However, during later gastrulation stages (8.5–10 hpf) tissue surface area and mediolateral (ML) width decreased rapidly (Fig. 1c, d) while tissue volume was largely preserved (Supplementary Fig. 1a). Notably, ML tissue width was reduced sequentially from anterior to posterior (Fig. 1b; Supplementary Fig. 1b) reminiscent of rotational movements leading to bending of the anterior-lateral tissue edges towards the dorsal midline. This tissue rearrangement culminated in an arrowhead-shaped tissue profile and formation of sharp neuroectoderm tissue boundaries at the end of gastrulation (10 hpf) (Fig. 1b). To corroborate whether ANP shape is indeed controlled by repositioning of neuroectoderm cells rather than modulation of gene expression and cell fate on/off states, we measured otx2:Venus expression levels over time in rearranging cells. We found that otx2:Venus signal increased in neuroectoderm fated cells over time compared to non-neuroectoderm cells outside of the ANP (Supplementary Fig. 1c), suggesting that the final shape of the ANP is defined through physical cell and tissue reorganisation rather than modulation of gene

expression. Collectively, these observations indicate that the early ANP is dramatically reshaped during late gastrula.

### Local cell internalisation drives global tissue flows to reshape the ANP

To understand how cellular processes within the ANP contribute to the observed global tissue shape changes, we evaluated morphogenetic processes on a cellular level in cell membrane (mRFP) labelled *Tg(otx2:Venus)* zebrafish embryos to image neuroectoderm cell movements and shape changes (Supplementary Movie 1). Quantitative analysis of cellular dynamics in curved 3D tissues, such as the neural plate, is an intricate problem and requires a robust methodology to transform 3D volumes into planar two-dimensional (2D) layers. To quantify local cellular processes across the entire ANP, we developed a computational image analysis workflow that enabled us to identify and align spatiotemporal landmarks in the embryo (ANP anterior edge and AP axis) and to compare cell and tissue dynamics across different imaging series and embryos (Supplementary Fig. 1d–f). This methodology also allowed us to visualise individual layers of the ANP tissue (superficial and deep cells) in the same plane by calculating planar projections of the whole 3D imaging volume (Supplementary Fig. 1g, h). This mapping process supported a downstream automated 2D image analysis of movements and segmentation of cells within the ANP (Supplementary Fig. 1i, j).

To quantify cell movements, we applied particle image velocimetry (PIV) to membrane labelled neuroectoderm cells in wild type (WT) embryos. Our analysis revealed the appearance of tissue flows with vortices on either side of the dorsal midline in the ANP at mid-gastrulation (8.5 hpf) (Fig. 1f), similar to flows previously identified in the early gastrula[32]. We further identified the presence of opposing cell movements along the dorsal midline of the embryo with posterior directed (rearward) and anterior directed (forward) flows that collided at a stagnation point located within the ANP tissue (~60 μm posterior of the ANP leading edge along the AP axis) (Fig. 1f, g). Rotational flows subsided after 8.75 hpf, at which time neuroectoderm cells displayed increasing mediolateral (ML) convergence movements towards the dorsal midline (Fig. 1f, h), which coincided with rapid narrowing of the tissue (Fig. 1b, d). Notably, lateral cells approaching the midline internalised and subsequently moved anteriorly, leading to a dorsal-to-ventral thickening (Fig. 1e, i). Collective internalisation along the ML axis was centred around the dorsal midline and occurred sequentially along the AP axis with the first appearance located posterior of the ANP leading edge (167 ± 9 μm s.e.m.; *n* = 3 embryos) (Fig. 1j). This was followed by a successive zipper-like anterior-to-posterior inward movement of neuroectoderm cells, suggesting that internalisation occurs stepwise along the dorsal midline rather than as a bulk movement (Fig. 1j; Supplementary Fig. 1k).

Given that the timing of neuroectoderm cell internalisation coincided with prominent changes in flow topologies, we hypothesised that internalisation may generate forces affecting tissue flows. To address this, we measured changes in orientation and shape of neuroectoderm cells moving from the anterior-lateral edges of the tissue to the dorsal midline. We found that the local orientation of cells (direction of major axis) dynamically aligned with the direction of the regional flow after internalisation and also became progressively stretched over time (Fig. 1k, l; Supplementary Fig. 1l). Taken together, these observations indicate that internalisation along the DV axis may generate forces that control large scale tissue flows along the AP/ML axes, enabling gradual symmetrical tissue reshaping (Fig. 1m).

### Differential behaviours of mesendoderm cells initiate ANP multilayer tissue folding and cell internalisation

Our observations indicate that opposing movements along the dorsal midline in the ANP initiate 3D reorganisation of the tissue. To investigate whether buckling and folding-like behaviour occurs in the tissue,

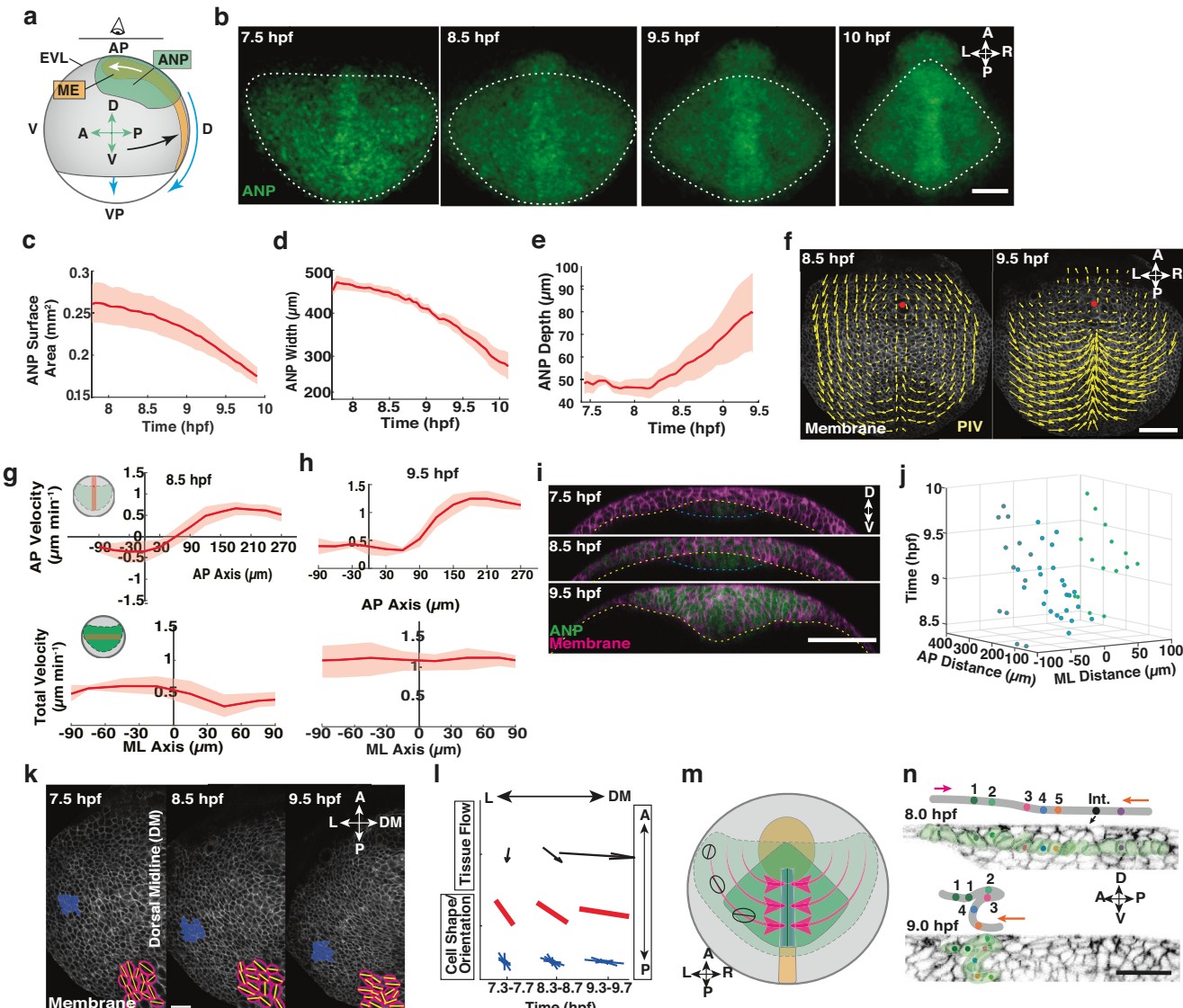

**Fig. 1 | Anterior neural plate reshaping during gastrulation is driven by global tissue flows. a** Schematic illustrating anterior neural plate (ANP, green), mesendoderm (ME, orange) and epithelial enveloping layer (EVL, grey) positions and movements in a zebrafish embryo during gastrulation (lateral view). Animal–vegetal pole (AP–VP) and dorsal–ventral (D–V) indicate embryo orientation. Green arrows: ANP tissue orientation, anterior (A), posterior (P), dorsal (D) and ventral (V). White arrow, ME movement; black arrow: convergent extension; blue arrow: epiboly movement. **b** Confocal images of ANP neuroectoderm (NE) cells in *Tg(otx2:Venus)* embryo during gastrulation. White dashes outline the ANP. Anterior, posterior, left (L), right (R) directions indicated. ANP surface area (**c**), width (**d**), and tissue depth (**e**) during gastrulation in wild type (WT) embryos. Time measured in hours post fertilisation (hpf). **f** Tissue flows of time-average velocities projected on membrane (mRFP) labelled NE cells in the dorsal-most layer of the ANP in WT embryos at 8.5 hpf and 9.5 hpf. Red dot: AP axis/ANP anterior edge intersection. Cell velocities from (**f**) at 8.5 (**g**) and 9.5 hpf (**h**) in AP direction along the dorsal midline (dm) and total velocities along mediolateral axis. AP velocity, X-axis: 0 = ANP anterior edge; negative anterior, positive posterior. Y-axis: negative posterior, positive anterior-directed flows. Total velocity, X-axis: 0 = dm; negative left, positive right. Insets: cell location in the embryo (red line). **i** Transverse confocal images (ventral view) of WT ANP through gastrulation. Cell membranes (mRFP, magenta) and ANP (otx2:Venus, green) labelled. Yellow dashes: NE and ME/yolk

interface. Blue dashes: ME ventral edge. **j** Time (hfp; z-axis) and location (xy-axes) of ANP cell internalisation (61 cells, 3 embryos). Blue dots: cells along dm, red dots and green dots cells on the right and left respectively. X-axis: distance from dm (0); Y-axis: distance posterior to ANP leading edge. **k** Membrane-labelled NE cells in the left half of the ANP during gastrulation. Orientation and shape of NE cells originating from the lateral leading edge of the ANP at 7.5 hpf (blue). Insets: cell outlines (red) as fitted ellipses and major cell axis (yellow segment). AP and LR axes indicated. **l** Tissue flow orientation and cell orientation/shape from (**k**). Arrows indicate average tissue flow direction over 20 min around time indicated, where size indicates magnitude. Average cell orientation and shape (length, red lines) and histogram of individual cell orientations (blue lines). **m** Schematic of ANP (green, AP view) shape changes and underlying ME (orange) during gastrulation. Pink arrow: flows towards dm. Cell orientation/shape indicated at different time points (AP view). **n** Representative confocal image series and illustrations showing tracked neuroectoderm cell positions over time within a single ANP tissue layer (green pseudocolour). Coloured dots indicate single cells. Purple arrow, posterior and orange arrow, anterior movements. Internalisation (Int) start point indicated. Orientation as in (**a**). All data are presented as the mean ± SEM of 3 embryos, unless stated otherwise. Scale bars: 100 μm (**b**, **f**, **i**); 50 μm (**n**). Source data are provided as a Source Data file.

we followed individual neuroectoderm cells at the ventral most layer along the mesendoderm interface during gastrulation. We found that neuroectoderm cells usually remain in contact with their respective neighbours and reorganise in such a way as to define a folded shape which orients parallel to the initial plane (Fig. 1n). Notably, multiple intertwined folds can be formed during internalisation, initiated by a smaller fold close to the ANP leading edge and followed by a prominent fold ahead of the internalisation start site that progressively moves anterior (Fig. 1n; Supplementary Fig. 1m). These finding suggest that deformations within the ANP bulk lead to asymmetric multilayer tissue folding which is distinct from symmetric folding in simple epithelia. Tissue folding can be driven by local mechanical instabilities which can be instigated locally by regional cellular actomyosin contractility[34–36]. Hence, we first investigated whether myosin 2 activity was enriched in cells at the dorsal side as observed at the midline internalisation site in the hindbrain[37], by staining for phosphorylated myosin 2 in neuroectoderm cells (8–9 hpf). Interestingly, we observed that the overall myosin 2 activity in neuroectoderm cells at or around the region of internalisation was overall low in comparison to clearly elevated phosphorylated myosin 2 levels in underlying axial mesendoderm cells (Supplementary Fig. 2a). Further, neuroectoderm cells that showed increased myosin 2 activity (mostly located at cell-cell junctions) compared to others, seemed randomly distributed throughout the ANP, with no clear correlation between elevated myosin 2 levels and pattern of internalising cells (Supplementary Fig. 2b).

We next investigated whether neuroectoderm cells undergo any shape changes once internalised (Supplementary Movie 2). When we quantified single cell deformations in 2D and 3D, we noticed that cells became progressively stretched with increasing cell depth in the direction of internalisation (Fig. 2a, b; Supplementary Fig. 2c, d). Interestingly, top and bottom surface areas of internalising cells were isotropically reduced during cell elongation (Supplementary Fig. 2e), suggesting that cell stretching is likely driven by ventral pulling forces or/and posterior pushing forces rather than cell intrinsic forces such as apical constriction.

To analyse such external forces that might locally trigger mechanical instabilities, we then mapped local tissue strain rates as readout of force-driven deformations[38]. Notably, we identified an area of localised, progressively increasing compression at the dorsal midline occurring before internalisation (7.5–8.25 hpf) (Fig. 2c, d; Supplementary Fig. 2f). This area matched the flow stagnation point along the AP axis (Fig. 1f, g), suggesting that opposing cell movements locally compact the tissue. We further observed that neuroectoderm cell internalisation initiated just posterior to the compressed area (Fig. 2c), indicating that tissue compaction provides a local resistance for the anterior motion of neuroectoderm cells that might cause a mechanical instability enabling internalisation (Supplementary Movie 2). Furthermore, our mediolateral (ML) strain analysis revealed that internalisation was accompanied by progressively increasing tissue compression on either side of the dorsal fold region (9.5 hpf; Fig. 2c, d), suggesting that internalisation along the DV axis may produce significant forces accounting for tissue convergence. Taken together, these measurements suggest that local deformations in the ANP likely result from multiple stresses acting in a spatiotemporally controlled manner.

We next focused on understanding how stresses leading to opposing cell movements and initiating multilayer folding might be generated in the ANP. To address this, we first analysed the behaviour of underlying axial mesendoderm progenitor (prechordal plate) cells along the tissue interface and determined the duration of adhesion between mesendoderm and overlying neuroectoderm cells (Supplementary Movie 2). Strikingly, we found that mesendoderm cells located at the front of the collective showed frequent contact exchanges with neuroectoderm cells, reminiscent of stick-slip motion typically observed in migrating cells on a substrate (Fig. 2e, f). In contrast, mesendoderm cells positioned at the rear of the collective and in the

posterior positioned notochord displayed long-lasting adhesions to overlying neuroectoderm cells (Fig. 2e, f). To analyse whether these differences could result from different degrees of cell motility, we measured the velocities of front and rear mesendoderm cells. We found that the velocities of rear cells increased over time and eventually exceeded those of front cell velocities (Supplementary Fig. 2g) This suggests that the observed spatial differences of binding duration between mesendoderm and neuroectoderm cells are not caused by cell motility differences, but rather result from discrete adhesive affinities of front and rear mesendoderm cells to the overlying ANP. We hypothesised that that these differences in adhesion could be explained by spatial variations of adhesion receptor recruitment between mesendoderm and neuroectoderm cells. We thus investigated the localisation of endogenous E-cadherin (cadherin 1, cdh1), which was previously shown to localise at the tissue interface between mesendoderm and neuroectoderm[32], by immunostaining. We found that cdh1 was more enriched between rear mesendoderm than front mesendoderm cells and neuroectoderm cells (Fig. 2g, h; Supplementary Fig. 3a–c) suggesting that increased cdh1 recruitment likely supports long lasting adhesions between mesendoderm rear and the overlying ANP.

We next explored whether spatial variations in cdh1 organisation specifically in the mesendoderm collective would explain such behaviours[32,39]. Thus, we generated a *Tg(gsc:cdh1-EGFP)* transgenic line which allowed live imaging of cdh1 dynamics specifically in the axial mesendoderm of embryos. Interestingly, we detected a greater accumulation of cdh1 in rear than front mesendoderm cells (Supplementary Fig. 3d, e) which was independent of promoter-driven expression levels (Supplementary Fig. 3f). Further, quantifying gsc:cdh1-EGFP (Supplementary Fig. 3g) or endogenous cdh1 (Fig. 2h) localisation at single cell-cell junctions at the interface between mesendoderm and neuroectoderm showed that cdh1 is more enriched between rear than front mesendoderm cells and overlying neuroectoderm. Collectively, these findings suggest differences in adhesion behaviour of front and rear mesendoderm cells with the overlying ANP.

To explore differences in migratory behaviour, we visualised F-actin processes in mesendoderm cells by imaging lifeact-EGFP dynamics (Supplementary Movie 3). We found that cells positioned at the front of the collective showed typically high protrusive activity at the neuroectoderm interface, while cells in the rear mostly lacked clear protrusion formation in the direction of migration (Fig. 2i; Supplementary Fig. 3h, i). Indeed, front cells displayed 6-fold higher protrusive activity than rear cells (Fig. 2i, j) and showed clear F-actin enrichment at sites of protrusions, whereas F-actin localisation in rear cells remained largely cortical (Fig. 2i, k). Together, these observations indicate that front mesendoderm cells resemble a highly polarised and migratory population whereas rear cells show a higher affinity to neuroectoderm cells by increased cdh1 recruitment to cell contacts at the interface.

Epiboly movements have been shown to be minimal near the animal pole[40], suggesting that any substantial movements in the ANP, which reaches the animal pole around 8 hpf[32], are presumably driven by adjacent mesendoderm cells. To further substantiate whether mesendoderm front cells might be responsible for posterior directed movements of neuroectoderm cells, we measured cell displacements around the animal pole including cells within and ahead (non-neuroectoderm) of the approaching ANP. In support of this hypothesis, we found that cells around the animal pole moved towards the approaching mesendoderm in a distance dependent manner (Supplementary Fig. 3j), suggesting that posterior directed neuroectoderm movements likely originate from the underlying mesendoderm. Together, these findings indicate that opposing movements in the ANP likely arise from differential migration and adhesion behaviours in the mesendoderm, whereby front cells exert posterior-directed pulling (traction) forces and rear cells drag neurectoderm cells anterior, in

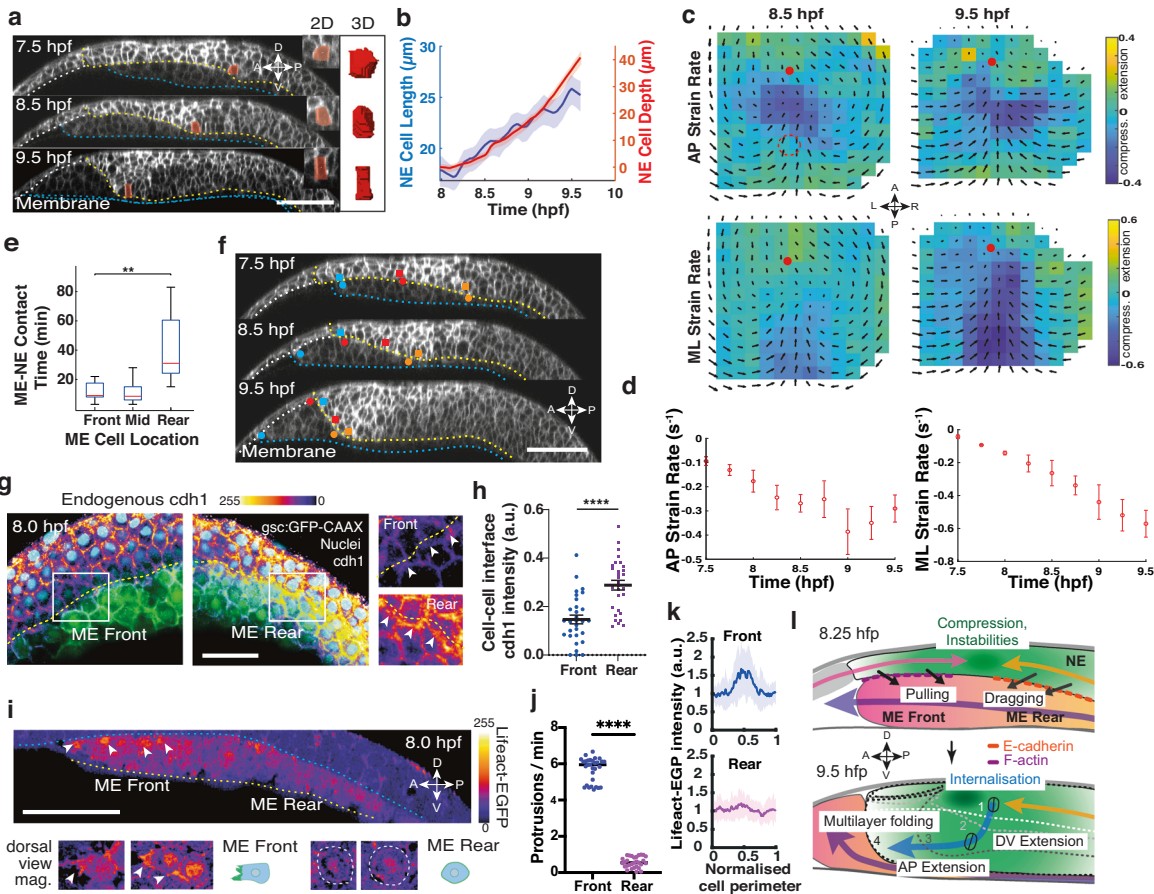

**Fig. 2 | Differential behaviours of mesendoderm cells initiate neuroectoderm internalisation. a** Sagittal confocal images of the anterior neural plate (ANP) in wild type (WT) embryos during gastrulation. Anterior (A), posterior (P), dorsal (D), ventral (V) indicated. Cell membranes (mRFP, white) labelled. Yellow dashes: neuroectoderm (NE) ventral and anterior mesendoderm (ME) border. Blue dashes: ME/yolk interface. White line: non-neuroectodermal tissue. Cell membranes (mRFP) in grey LUT. Representative internalising cell (red) illustrates shape change in 2D and 3D over time. **b** Internalising NE cell depth (red) and lateral cell length (blue) during gastrulation (42 cells, 3 embryos). **c** Average ANP strain rates along the AP (upper) and mediolateral (ML, lower) axes of WT embryos. Minimum, green (0); maximum stretching, yellow; maximum compression, blue. Black arrows, average tissue flows. Red dot: AP axis/ANP leading edge intersection. Red dashed circle: initial position of internalisation. **d** Maximum absolute ANP strain rates along the AP (left) and ML (right) axes of WT embryos during gastrulation. Negative values indicate compression. **e** Contact time between ME and NE cells [locations marked in (f)] along the AP axis (9 cells, 3 embryos). Kruskal–Wallis test (**, $p < 0.01$). Central line indicates the median, bottom and top edges of the box indicate the 25th and 75th percentiles, respectively. Whiskers extend to the most extreme data points not considered outliers. **f** Sagittal confocal images showing evolution of ME (circles) and NE (squares) cells at the ME front (blue), mid (red), and rear (orange) during gastrulation. Orientation and labels as in (**a**). **g** Sagittal confocal images showing cell-cell contacts between front and rear ME and NE stained for endogenous cdh1 (Fire LUT; white maximum, black minimum), nuclei (cyan) and native gsc:GFP-CAAX signal. Boxed areas are magnified to the right

highlighting cdh1 localisation at the front and rear. White arrowheads indicate contacts between ME cells and overlying NE cells. Yellow line demarks interface (slight offset). **h** Quantification of endogenous cdh1 localisation at cell-cell contacts between NE and ME cells in the front (blue circles) and rear (purple squares) of the collective in a WT embryo. Data is presented as individual normalised intensity values (30 cells, 3 embryos). Two-tailed unpaired $t$-test (****, $p = 2e^{-6}$). **i** Sagittal (xz) multiphoton image of *Tg(actb1:lifeact-EGFP)* embryo showing F-actin localisation (Fire-LUT; white maximum, black minimum). White arrowheads indicate F-actin rich protrusions at the ME front. Below, magnified images (xy) and schematics of cells at the ME front with polarised F-actin rich protrusions and ME rear with cortical F-actin localisation. **j** Number of total cell protrusions observed over time in the front (blue) and rear (purple) of *Tg(actb1:lifeact-EGFP)* embryos (20 cells each, 3 embryos). Two-sided Mann Whitney test (****, $p < 0.0001$).
**k** Normalised intensity of F-actin (lifeact-EGFP) measured along the cell periphery (0.5 aligned with direction of collective motion) within single front (blue) and rear (purple) ME cells adjacent to the NE interface (10 cells each, 3 embryos).
**l** Schematic (lateral view) illustrating forces (arrows) exerted by the migrating ME (pink/orange) leading to opposing movements (pink arrow posterior, orange arrow anterior) in the ANP (green), generating mechanical instabilities resulting in NE internalisation and multilayer folding and subsequent tissue extension along the DV and AP axes. Orange dashes: E-cadherin. Purple dashes: F-actin. Cell shapes depicted. All data are presented as the mean ± SEM of 3 embryos, unless otherwise stated. Scale bars: 100 μm. Source data are provided as a Source Data file.

agreement with recently reported frictional forces acting between these tissues[32]. We propose that an interplay of these forces is sufficient to account for local mechanical instabilities in the ANP to initiate cell internalisation and multilayer tissue folding (Fig. 2l).

### Internalisation depends on E-cadherin-mediated tissue interaction
We thus sought to identify how the hypothesised mechanical coupling between the neuroectoderm and mesendoderm enabling cell

internalisation is established on a molecular level. Recent reports highlighted important roles for E-cadherin (cadherin-1) as well as for N-cadherin (cadherin-2) in mechanical interactions between the zebrafish neural plate and the underlying mesendoderm[32,37]. To test for a functional role of both of these adhesion receptors in regulating internalisation of the neuroectoderm, we knocked-down the levels of both adhesion molecules individually. We found that ANP shape was not altered during gastrulation in N-cadherin depleted embryos (*cdh2* morphants; *cdh2*) (Supplementary Fig. 4a–c). Further, *cdh2* morphants

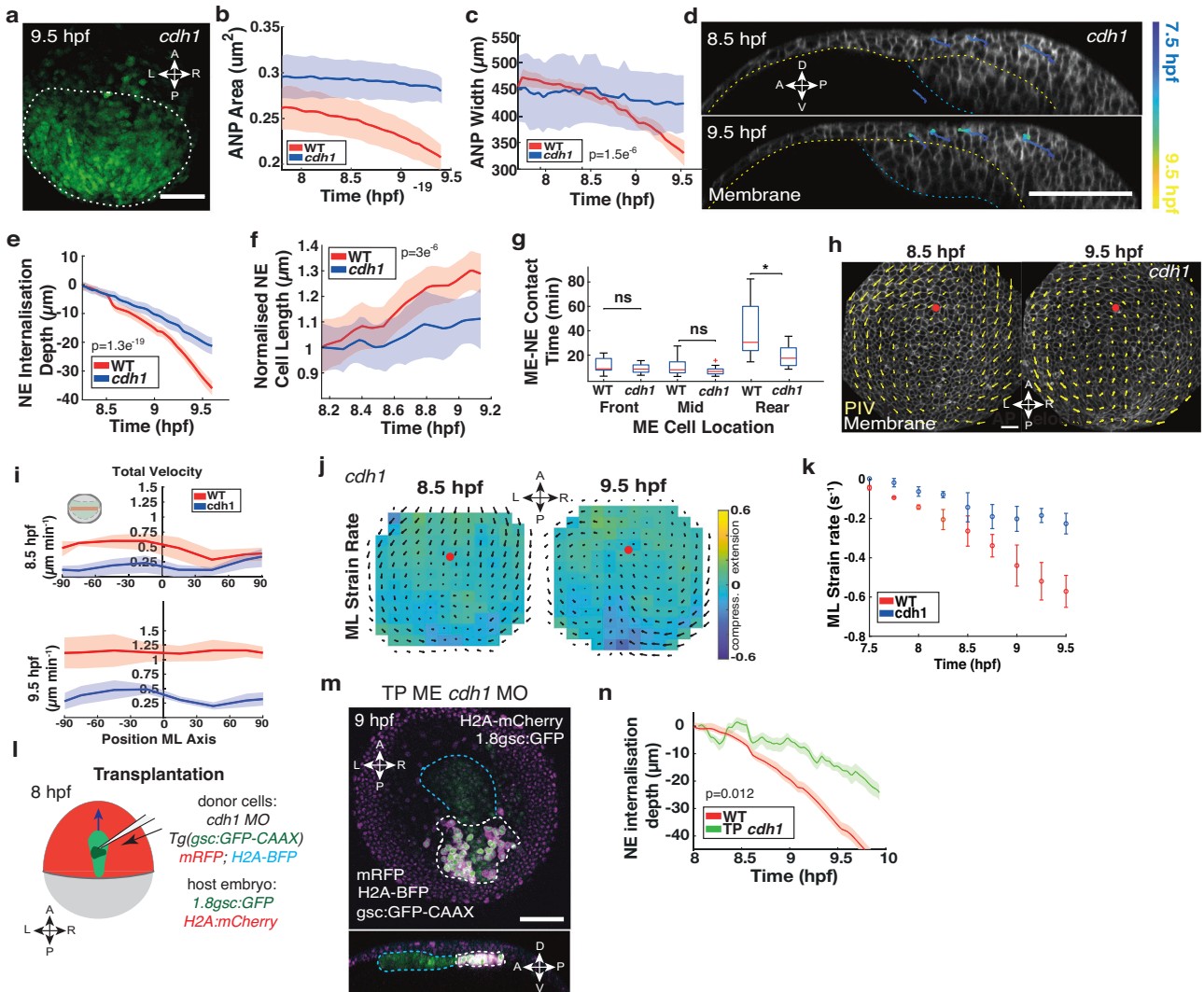

**Fig. 3 | E-cadherin-mediated inter-tissue adhesion is required for neuroectoderm internalisation. a** Confocal imaging of neuroectoderm (NE) cells in the anterior neural plate (ANP) of *Tg(otx2:Venus) cdh1* morphant embryos during gastrulation. White dashes outline the ANP. Anterior (A), posterior (P), left (L), and right (R) indicated. ANP surface area (**b**) and width (**c**) of WT (red) and *cdh1* morphant (blue) embryos during gastrulation. One-way Kruskal–Wallis test ($p = 1.5e^{-6}$). **d** Sagittal confocal imaging of *cdh1* morphants with representative cells tracked within the ANP through gastrulation. Cell membrane (mRFP) labelled white. Yellow dashes: NE/mesendoderm (ME) border. Blue dashes: ME/yolk interface. Cell tracks coloured blue (7.5 hpf) to yellow (9.5 hpf). **e** NE internalisation depth in WT and *cdh1* morphants during gastrulation (30 cells, 3 embryos. One-way Kruskal–Wallis test ($p = 1.3e^{-19}$). **f** Internalising NE cell length in WT and *cdh1* morphants during gastrulation (30 cells, 3 embryos). Two-sided unpaired *t*-test ($p = 3e^{-6}$). **g** Contact duration between NE and mesendoderm (ME) cells along the AP axis in WT and *cdh1* morphants (9 cells, 3 embryos). Central line indicates the median, bottom and top edges of the box indicate the 25th and 75th percentiles, respectively. Whiskers extend to the most extreme data points not considered outliers. + marker symbol indicates an outlier. Two-tailed unpaired *t*-test (front $p = 0.27$; mid $p = 0.12$; rear *: $p = 0.04$). **h** Tissue flows of time-averaged velocities projected on membrane

(mRFP) labelled NE cells in the dorsal-most layer of the ANP in *cdh1* morphants at 8.5 hpf and 9.5 hpf. Red dot: AP axis/ANP anterior edge intersection. **i** Total cell velocities from (**h**) along the mediolateral (ML) axis of WT and *cdh1* morphants at 8.5 hpf and 9.5 hpf. X-axis: 0 marks dorsal midline; negative left, positive right. **j** Time-averaged ANP domain strain rates along the ML axes of *cdh1* morphants. Minimum green (0); maximum stretching yellow, maximum compression blue. Black arrows: time-averaged tissue flows. Red dot: AP axis/ANP leading edge intersection. **k** Maximum absolute strain rates along the ML axis of WT and *cdh1* morphants during gastrulation. Negative values represent compression. **l** Transplantation schematic. Donor *cdh1* morpholino (MO) expressing mesendoderm cells (dark green) transplanted in the rear of WT host mesendoderm (light green) before internalisation at 8 hpf. **m** Representative multiphoton image showing transplanted (TP) *cdh1* MO expressing ME cells (mRFP; H2A-BFP; gsc:EGFP-CAAX; white dashed line) at the ME rear in a host embryo (H2A:mCherry; 1.8gsc:GFP; blue dashed line) at 9 hpf. Upper panel, animal pole view and lower panel, sagittal section. **n** NE internalisation depth of dorsal NE layer in WT and *cdh1* MO transplanted embryos (2 embryos, 5 cells each). Two-sided Wilcoxon rank sum test, ($p = 0.012$). All data are presented as the mean ± SEM of 3 embryos, unless otherwise stated. Scale bars: 100 μm. Source data are provided as a Source Data file.

displayed similar neuroectoderm internalisation behaviour to WT (Supplementary Fig. 4d, e), indicating that N-cadherin is likely dispensable for shaping the ANP tissue during gastrulation.

We next moderately reduced the amount of E-cadherin in embryos (*cdh1* morphants; *cdh1*) to avoid early deep cell epiboly arrest, and to allow for sufficient development until the end of gastrulation[41,42]. Live imaging of *cdh1* morphants (Supplementary

Movie 4) revealed very little change in ANP surface area or width suggesting strongly diminished ANP tissue remodelling (Fig. 3a–c; Supplementary Fig. 4f, g). Remarkably, neuroectoderm cells did not internalise and remained in superficial layers in these embryos (Fig. 3d, e; Supplementary Fig. 4h) with a strongly reduced tendency to elongate in the DV direction as observed in WT embryos (Fig. 3f; Supplementary Fig. 4i). We hypothesised that changes in inter-tissue

force coupling should primarily affect mesendoderm cells at the rear responsible for dragging neuroectoderm cells anterior and measured the duration of heterotypic tissue adhesion. Indeed, we found that specifically rear mesendoderm adhesion to neuroectoderm was significantly reduced in *cdh1* morphants (Fig. 3g). While opposing movements along the dorsal midline were still detectable in *cdh1* morphants at 8.5 hpf, velocities remained substantially low in the tissue throughout gastrulation compared to WT (Fig. 3h, i; Supplementary Fig. 4j). Further, local tissue compaction along the AP axis required for initiation of ANP internalisation at 8.5 hpf was absent compared to WT (Supplementary Fig. 4k, l), as well as ML tissue deformations along the dorsal midline were strongly reduced at 9.5 hpf (Fig. 3j, k).

To further substantiate our observation, we locally weakened E-cadherin adhesion between the two tissues by transplanting *cdh1* MO mesendoderm cells specifically into the rear of the collective before internalisation (8 hpf) (Fig. 3l, m, Supplementary Fig. 4m, n). Strikingly, decreasing local interactions between the mesendoderm rear and overlying neuroectoderm in this way was sufficient to interfere with internalisation and neuroectoderm cells remained more superficial (Fig. 3n; Supplementary Fig. 4o), which reinforces a substantial role for E-cadherin in regulating this process. Taken together, our results suggest that E-cadherin, but not N-cadherin, mediated adhesion is critical to enable neuroectoderm internalisation by locally supporting inter-tissue force coupling.

## A spatiotemporal interplay of forces predicts global tissue flows in the ANP

Our observation that global flow patterns change during gastrulation suggests that neuroectoderm tissue flows are regulated by multiple forces that vary in space and time. To address this, we developed a mechanical model of neuroectoderm tissue flows to determine the minimal set of local forces required to reproduce tissue flows during different gastrulation stages (8.25–10 hpf) (Fig. 4a–j). On symmetrising (i.e., left-right averaging) experimental flow data, we found that flows before internalisation (8.25 hpf) exhibited vortices on either side of the dorsal midline and a nearby extension point on the midline (Fig. 4a). After the initiation of internalisation (8.75 hpf), the symmetrised flows changed and displayed a turning flow on either side of the midline and a sink on the midline that is the signature of internalised cells leaving the plane of the ANP (Fig. 4f). To reproduce these flow topologies *qualitatively*, we modelled the neuroectoderm as an infinite two-dimensional incompressible viscous fluid, in which flows are driven by point force and point sink singularities[43] on an axis representing the dorsal midline (Supplemental Note).

First, we asked whether forces acting on the neuroectoderm could reproduce the flow topology observed in vivo before internalisation (8.25 hpf) (Fig. 4b–e). We found that neither a single force singularity, nor two parallel ones could account for the observed topological features (Fig. 4b, c). However, two opposing force singularities acting on the neuroectoderm along the dorsal midline were able to account for the observed tissue flows before internalisation (Fig. 4d, e). Strikingly, this model predicts that the anterior-directed force must be stronger than the posterior-directed one to reproduce the observed orientation of the vortices (Fig. 4e). Next, we modelled the flows after initiation of internalisation (8.75 hpf) (Fig. 4g–j). To represent forces directed out of the ANP tissue plane that led to internalisation of cells at the midline, we introduced a single sink singularity on the AP axis. We found that a sink on its own (Fig. 4g), or in combination with a force directed towards the posterior (Fig. 4h) could not reproduce the experimental flow topology (Fig. 4f). However, combinations of a sink and a single force directed towards the anterior (Fig. 4i) or two opposing forces (Fig. 4j) reproduced this topology.

Our qualitative model thus indicates two mechanical regimes: Flows before tissue internalisation result from opposing forces along the AP tissue axis, while an anterior-directed force and an internalisation force out of the tissue plane drive flows during internalisation. To test this hypothesis, we sought to show that the minimal combination of two forces and one sink that we have shown above to be *necessary* to reproduce the flow topologies *qualitatively*, is in fact sufficient to reproduce the experimentally measured flows *quantitatively*. For this purpose, we extended our model (Supplemental Note) by including friction between the neuroectoderm and the overlying enveloping layer (EVL) to remove the Stokes paradox[43] and regularising the divergences of the force and sink singularities, smearing them out over finite distances that represent the characteristic physical extents over which they are applied to the tissue. We thus fitted the positions, magnitudes, directions, and extents of two such regularised forces and one such regularised sink and the EVL-neuroectoderm friction to the experimentally quantified tissue flows for different timepoints (8.25–10 hpf) (Fig. 4k–o and Supplemental Note).

Importantly, this minimal mechanical model could indeed quantitatively reproduce the experimentally observed neuroectoderm tissue flows (Fig. 4k, l). The fits revealed multiple mechanical transitions during gastrulation (8.25–10 hpf) (Fig. 4m–o) and in particular differences in the directions, relative positions, and magnitudes of the forces and sink driving the flows (Fig. 4m–o). Fits for early time points before internalisation (<8.5 hpf) featured two opposing forces along the AP axis (the smaller posterior-directed, the larger anterior-directed) (Fig. 4k, m), agreeing with predictions from our qualitative model (Fig. 4a–e). The model predicted a first mechanical transition when internalisation begins (~8.5 hpf) and the two opposing forces gave way to two parallel forces directed towards the anterior (Fig. 4m), with a simultaneous increase in EVL-neuroectoderm friction (Fig. 4o). Consistently with the start of internalisation, the fitted sink velocity increased (Fig. 4n) and reached a maximum during a second mechanical transition during late gastrulation (~9 hpf) (Fig. 4n). Interestingly, forces repositioned towards the leading edge of the tissue at that time (Fig. 4m), indicating positional dynamics of the exerted forces. During late gastrulation (~10 hpf), the fitted sink strengths decreased, and the forces and friction were considerably reduced (Fig. 4m, o), suggesting a final mechanical relaxation as the ANP acquires its final shape. In summary, we have shown that the minimal mechanics identified by our qualitative model (Fig. 4a–j) are sufficient to reproduce the experimental flows quantitatively (Fig. 4k, l). Moreover, our quantitative model revealed how mechanical transitions at different developmental stages lead to a spatiotemporal force distribution regulating the experimentally measured flows.

## Mesendoderm migration is essential for neuroectoderm cell internalisation

Based on our in vivo and in silico analyses, we hypothesised that the underlying mesendoderm may act as a key force generator driving ANP shape changes. To substantiate this mechanical hypothesis further, we investigated ANP morphogenesis in the absence of mesendoderm cells and imaged neuroectoderm cell dynamics in *lefty1* mRNA injected *Tg(oxt2:Venus)* zebrafish embryos (Supplementary Movie 5). We observed that, in these embryos, the ANP was misshaped and remained close to the embryonic margin (Fig. 5a; Supplementary Fig. 5a), consistent with previous observations of fixed tissue staining in MZ*oep* mutant embryos[32,44]. In contrast to WT embryos, we observed that the tissue width was nearly unaltered in *lefty1* embryos during gastrulation (Fig. 5b; Supplementary Fig. 5b). Importantly, we detected no characteristic internalisation of neuroectoderm cells (Fig. 5c, d; Supplementary Fig. 5c), or rotational flow patterns (Fig. 5e) in embryos lacking mesendoderm. Further, *lefty1* embryos showed wide-ranging ANP tissue compression along the AP axis without significant ML deformations, driven by unidirectional posterior flows of neuroectoderm cells against the margin (Fig. 5f, g; Supplementary Fig. 5d, e). Consequently, neuroectoderm cells became oriented perpendicular to the tissue flow along the AP tissue axis (Supplementary Fig. 5f).

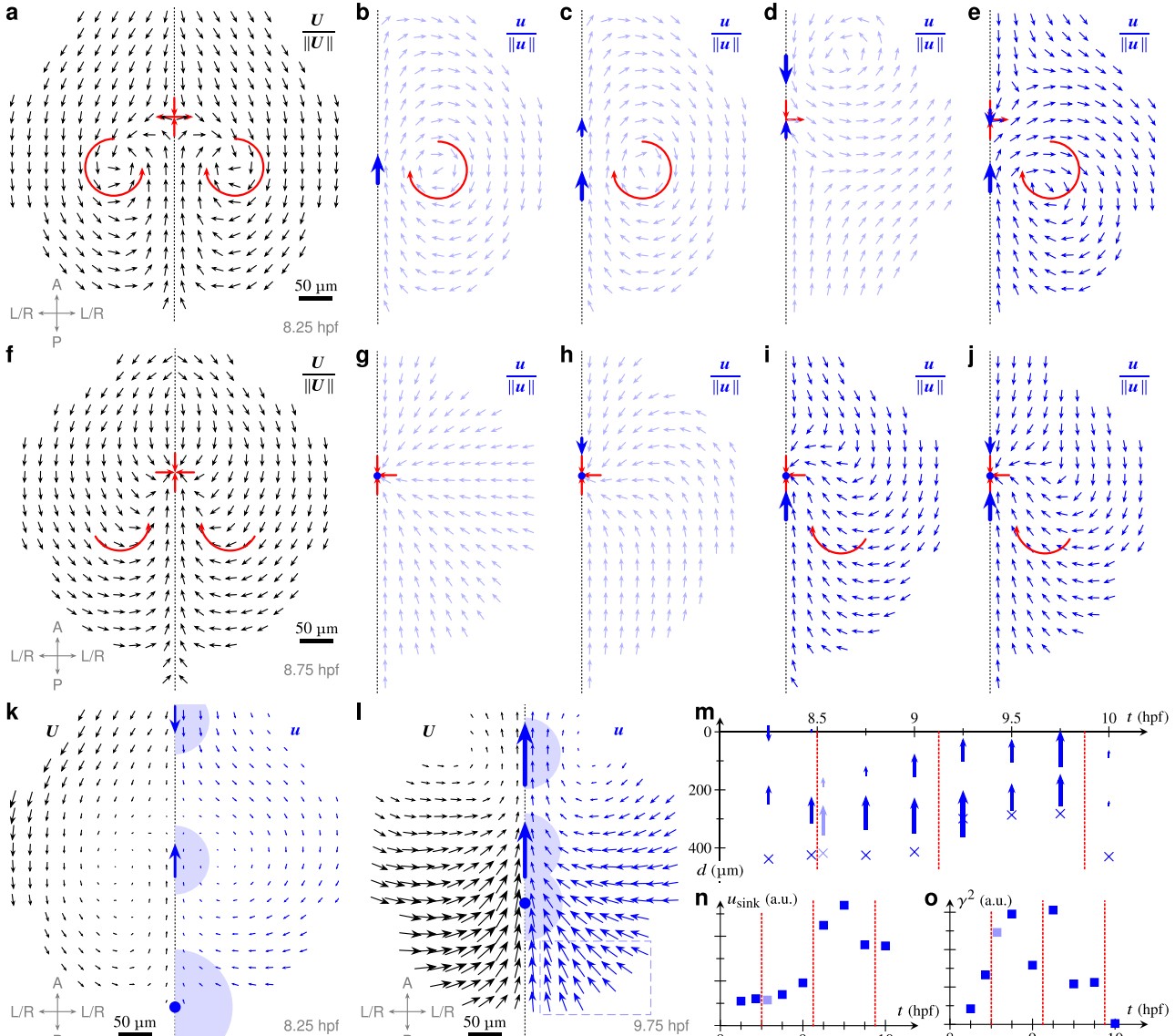

**Fig. 4 | Mechanical modelling predicts spatiotemporal interplay between forces controlling tissue flows in the anterior neural plate. a** Symmetrised experimental flow directions before internalisation (8.25 hpf); flow features lateral vortices and an extension point along the dorsal midline (red). Inset: embryo axes (A: anterior, P: posterior, L/R: left-right symmetrised). **b** Flow directions from a single, anterior directed force. Flow features a lateral vortex but no extension point. **c** Flow directions from two parallel force singularities; flow features a lateral vortex but no extension point. **d, e** Flow directions from two antiparallel force singularities. **d** Posterior-directed force dominates. Flow displays an extension point but incorrect vortices. **e** Anterior-directed force dominates. Extension point and vortices are correct. **f** Symmetrised experimental flow directions after internalisation initiation (8.75 hpf). Flow features an axis sink and a turning flow posterior to the sink. Vortices are lost. **g** Flow directions from a single sink lack the turning flow. **h** Flow directions from a sink and a pulling force; turning flow orientation is incorrect. **i** Flow directions from a sink and a drag force feature a sink and a turning

flow with correct orientation. **j** Flow directions from a sink, drag force, and a smaller pulling force feature a sink and correct turning flow. **k** Experimental flow field $U$ (left) and fitted flow field $u$ (right) at 8.5 hpf. Arrows indicate force directions and positions; blue dot indicates sink position. Shading indicates areas over which the singularities are smeared out. Fitted sinks represent contributions from compressibility and internalisation. **l** Experimental and fitted flow fields at 9.75 hpf. Blue dashed area highlights underestimated left-right flows in the posterior region. **m** Positions of the regularised force singularities (arrows) and regularised sinks (circle) against time, given in terms of the distance $d$ from the anterior limit of the tissue. Arrow size: force magnitude. Different fits of very similar fit scores shown for 8.5 hpf. Vertical red lines separate different mechanical regimes. **n** Normalised fitted sink velocity ($u_{sink}$) against time. Red dashes: mechanical transitions. AU arbitrary units. **o** Normalised fitted friction coefficient ($\gamma^2$) against time. Red dashes: mechanical transitions. All scale bars, 50 µm.

Together, this indicated that axial mesendoderm migration provides essential extrinsic forces to regulate internalisation and tissue flows to reshape the ANP during gastrulation.

We further reasoned that blocking mesendoderm migration in WT embryos may eliminate the anterior drag of neuroectoderm cells and therefore abolish the generation of instabilities required for internalisation. To address this, we inhibited the small Rho GTPase Rac1 in mesendoderm cells, an essential protein for protrusion

formation in mesendoderm cells[45]. We locally transplanted mesendoderm dominant negative Rac1 (dnRac) expressing or WT control cells from donor embryos to the leading edge of mesendoderm in WT host embryos before neuroectoderm internalisation occurred (7.5–8 hpf) (Fig. 5h, i; Supplementary Fig. 5g, h). Notably, dnRac expressing cells displayed very few protrusions compared to transplanted control cells (Supplementary Fig. 5i, j) and stalled anterior migration of endogenous mesendoderm cells compared to transplanted control cells

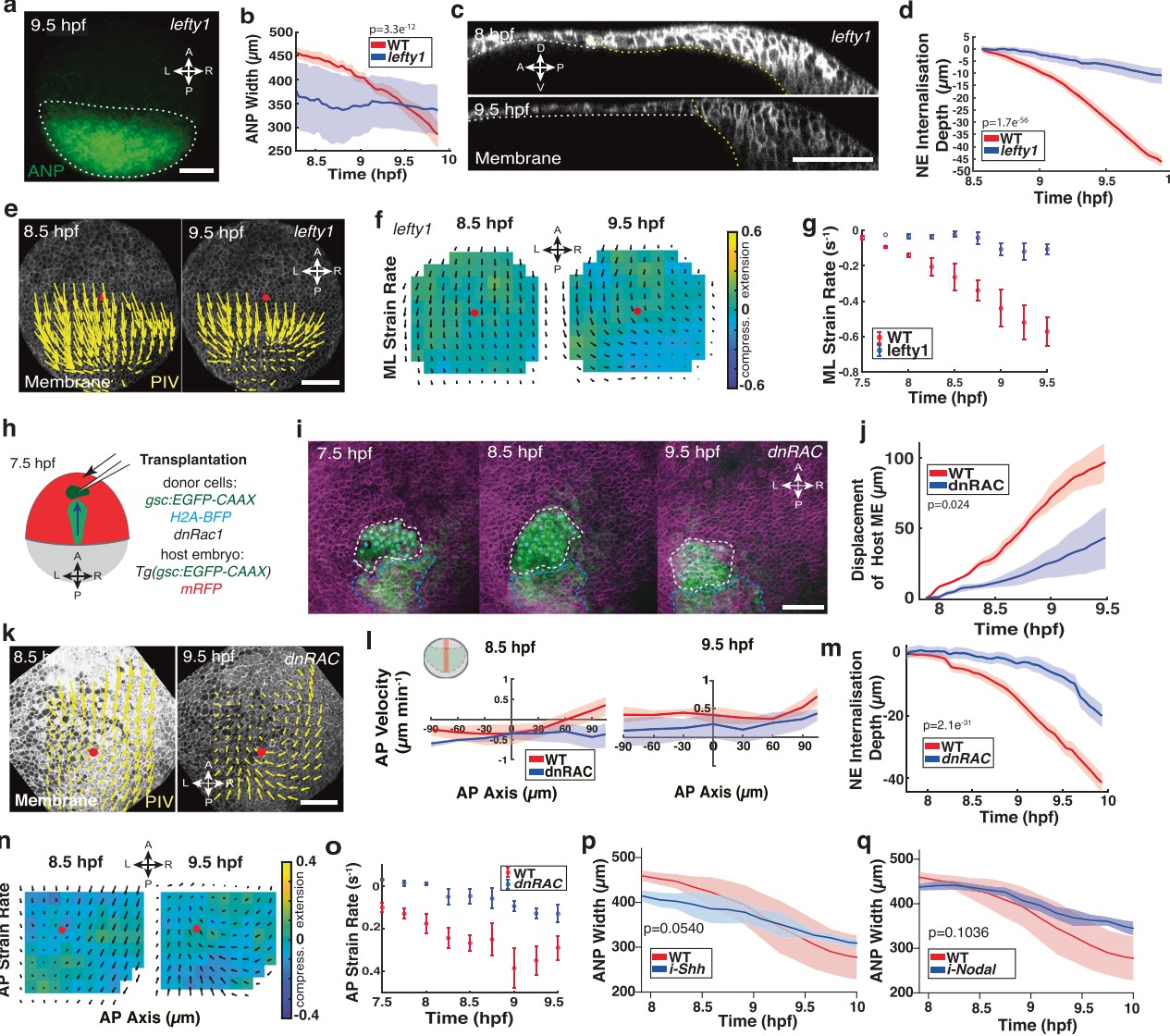

**Fig. 5 | Mesendoderm migration is essential for anterior neural plate internalisation and multilayer folding. a** Confocal images of neuroectoderm (NE) cells in the anterior neural plate (ANP) of *Tg(otx2:Venus) lefty 1* morphants at 9.5 hpf. White dashes outline the ANP. **b** ANP width in WT (red) and *lefty 1* (blue) morphants during gastrulation. Two-sided unpaired *t*-test ($p = 3.3e^{-12}$). **c** Sagittal confocal images of the ANP in *lefty 1* morphants at 8 hpf and 9.5 hpf. Dorsal (D), ventral (V), anterior (A), posterior (P) indicated. Cell membranes (mRFP) grey LUT. Yellow dashes, NE/yolk and white dashes, non-NE/yolk interface, respectively. **d** Internalisation depth of NE cells in WT and *lefty 1* morphants during gastrulation (30 cells, 3 embryos). Two-sided unpaired *t*-test ($p = 1.7e^{-56}$). **e** Average tissue flow velocities projected on confocal imaging of membrane labelled NE cells in the dorsal-most layer of the ANP in *lefty 1* morphants at 8.5 hpf and 9.5 hpf. Red dot: AP axis/ANP anterior edge intersection. **f** Average ANP domain strain rates along the mediolateral (ML) axis of *lefty 1* morphants. Strain rate minimum green, maximum stretching yellow, maximum compression blue. Black arrows: average tissue flows. Red dot: AP axis/ANP leading edge intersection. **g** Maximum absolute strain rates along ML axis of WT and *lefty 1* morphants during gastrulation. Negative values indicate compression. **h** Transplantation schematic. Donor *dnRac1* expressing ME cells (dark green) transplanted anterior to host ME (light green) before internalisation at 7.5 hpf. **i**) Confocal images (AP view) of *Tg(gsc:EGFP-CAAX)* host embryo (mRFP) transplanted with *dnRac1* expressing ME donor cells (*gsc:GFP-*

*CAAX*, H2A-BFP, white dashes), anterior to host ME (blue dashes) at 7.5 hpf and imaged throughout gastrulation (7.5–9.5 hfp). **j** Anterior displacement of mesendoderm in WT (red) and *dnRac1* transplanted (blue) embryos. Two-sided unpaired *t*-test ($p = 0.024$). **k** Tissue flow of time-average velocities projected on membrane labelled NE cells in the dorsal-most layer of the ANP in transplanted embryos at 8.5 hpf and 9.5 hpf. Red dot: AP axis/ANP anterior edge intersection. **l** Cell velocities from (**k**) at 8.5 hpf and 9.5 hpf in AP direction along the dorsal midline of WT and transplanted embryos. AP velocity plots, X-axis: 0 = ANP anterior edge; negative anterior, positive posterior. Y-axis: negative posterior, positive anterior-directed flows. **m** NE internalisation depth of dorsal NE layer in WT and *dnRac1* transplanted embryos. Two-sided unpaired *t*-test ($p = 2.1e^{-31}$). **n** Average ANP domain strain rates along the AP axis of WT and *dnRac1* transplanted embryos. Minimum green (0); maximum stretching yellow; maximum compression blue. Black arrows: average tissue flows. Red dot: AP axis/ANP leading edge intersections. **o** Maximum absolute ANP strain rates along the AP axis of WT and *dnRac1* transplanted embryos during gastrulation. Negative values indicate compression. **p** ANP width in WT (red) and Shh inhibited (blue) embryos during gastrulation. Two-sided unpaired *t*-test ($p = 0.0540$). **q** ANP width in WT (red) and Nodal inhibited (blue) embryos during gastrulation. Two-sided unpaired *t*-test ($p = 0.1036$). All data are presented as the mean ± SEM of 3 embryos, unless otherwise stated. Scale bars: 100 μm. Source data are provided as a Source Data file.

(Fig. 5j; Supplementary Fig. 5k, l, m). Consequently, on transplantation of dnRac expressing cells, the neuroectoderm only displayed unidirectional posterior-directed flows along the dorsal midline at 8.5 hpf, suggesting that forces controlling anterior-directed neuroectoderm movements were abolished (Fig. 5k, l; Supplementary Fig. 5n). Notably, we did not observe any internalisation when transplanted dnRac expressing cells interfered with mesendoderm migration (Fig. 5m; Supplementary Fig. 5k, o), while internalisation occurred normally in control cell transplantations (Supplementary Fig. 5p). Strain maps revealed that stresses in the ANP were altered and local AP tissue compaction at 8.5 hpf as well as lateral compression due to internalisation at 9.5 hpf were strongly reduced (Fig. 5n, o; Supplementary Fig. 5q, r). However, we noticed that internalisation movements were partially restored by late gastrulation (9.5 hpf) (Fig. 5m; Supplementary Fig. 5k) when the mesendoderm motion resumed (Fig. 5j), likely through AP axis extension of posterior tissues. Overall, we conclude that mesendoderm migration during gastrulation constitutes a major force generating mechanism providing extrinsic forces critical for the correct timing of cell internalisation.

In order to identify a molecular signalling pathway that might alternatively regulate cell internalisation, we tested whether morphogens previously implicated in neural plate development, may impact on ANP formation during gastrulation. We first explored a potential role for Sonic hedgehog (Shh), which is produced in the axial mesendoderm and has been shown to be important for DV neural plate patterning[46]. We found that shaping of the ANP in Shh inhibited embryos occurred similar to WT embryos and did not affect neuroectoderm internalisation behaviour (Fig. 5p and Supplementary Fig. 6a, b), suggesting a negligible role for Shh in remodelling the ANP during gastrulation. Next, we investigated a potential dual role for Nodal signalling independent of its role in fate specification. Recent studies demonstrated a function for Nodal in supporting neuroectoderm morphogenesis[47] and in modulating E-cadherin dynamics in mesendoderm cells[48]. When we inhibited Nodal signalling in embryos during gastrulation, we found that neuroectoderm cell internalisation occurred normally (Supplementary Fig. 6c, d). Interestingly, ANP tissue reshaping did not complete entirely in those embryos and the tissue remained laterally extended at later stages of gastrulation (Fig. 5q) which is likely linked to the role for Nodal signalling in promoting convergence and extension in the neuroectoderm[47]. Collectively, these observations further strengthen the idea that internalisation in the ANP is regulated mechanically by differential cell interactions with the underlying mesendoderm.

**Convergent extension movements are dispensable for internalisation but required for ANP tissue extension**
Our in silico analysis revealed a second anterior directed force acting at the posterior end of the ANP during late gastrulation (Fig. 4l). We hypothesised that this force may be associated with convergent extension (CE) movements which are highly conserved global cell movements essential for embryonic axis formation. CE was shown to be responsible for the generation of tissue-scale force production driving narrowing and stretching of posterior mesendoderm structures such as the notochord[49,50]. To address how CE contributes to ANP tissue morphogenesis, we generated *Tg(otx2:Venus); wnt11f2* (*wnt11*)[51] transgenic mutants to image ANP formation in embryos lacking CE movements (Supplementary Movie 6). Measurements of live tissue dynamics in these embryos revealed that the ANP is only partially reshaped during gastrulation and remains laterally expanded compared to WT embryos (Fig. 6a, b; Supplementary Fig. 7a, b). We next analysed cellular rearrangements within the neuroectoderm to identify whether changes in internalisation and/or tissue flows could account for defective ANP shape. Strikingly, we found that the timing and spatial location of initiation of internalisation are largely preserved in *wnt11* mutant embryos (distance posterior of ANP leading edge:

*wnt11*: 181 ± 12 μm s.e.m; WT: 167 ± 9 μm s.e.m.; *n* = 3 embryos each) (Fig. 6c), regardless of the slight decrease in mesendoderm velocity (Supplementary Fig. 7c, d). However, internalisation in *wnt11* mutants showed an increased neuroectoderm internalisation depth (Fig. 6d; Supplementary Fig. 7e). Furthermore, the range of anterior displacement of the internalised tissue was drastically reduced and remained further posterior throughout gastrulation compared to WT embryos (Fig. 6e), suggesting that CE supports AP extension of the ventral ANP tissue. To test whether this limited range of internalised tissue motion would affect force generation, we analysed neuroectoderm tissue flows in *wnt11* mutant embryos. We found that flow topologies and cell orientations along the flow in *wnt11* mutants were qualitatively similar to the WT at 8.5 hpf (Fig. 6f; Supplementary Fig. 7f, g). However, flow velocities along the AP, and especially the ML axis, were considerably reduced in stages following internalisation (9.5 hpf; Fig. 6f, g). Remarkably, we found that tissue deformations and strain rates in *wnt11* mutant embryos were comparable to the WT at early stages (8.5 hpf) but reduced at later stages (9.5 hpf) (Fig. 6h, i; Supplementary Fig. 7h, i). Neuroectoderm cell internalisation remained locally restricted to the posterior of the ANP in *wnt11* mutants (Fig. 6c) and lacked the typical internalisation pattern along the AP axis observed in WT embryos (Supplementary Fig. 1k). Interestingly, *wnt11* mutant embryos displayed a kink in the ANP on the dorsal site of internalisation (Supplementary Fig. 7j), indicating that ventral tissue extension supported by CE is integral in preserving proper ANP tissue geometry. Collectively, we conclude that CE movements are dispensable for internalisation but essential to maintain robust tissue flows and ANP reshaping by mechanically supporting anterior tissue extension following internalisation.

## Discussion
Our findings highlight a new mechanical mechanism consisting of multiple sequential overlapping steps (multi-tiered) essential for regulating global morphogenetic flows in the ANP by a coordinated spatiotemporal-dependent interplay of extrinsic forces and mechanical tissue coupling (Supplementary Fig. 8). Inter-tissue adhesion and force coupling are thus emerging as key processes in shaping developing embryos[16,32,52–54]. We propose that tissue flows in the ANP are critical to regulate the final tissue shape at the end of gastrulation preceding neurulation. Tissue flows were previously shown to depend on an interplay of multiple tissue intrinsic and/or extrinsic processes which can provide forces necessary to drive oriented tissue flows in the embryo[55–60]. Interestingly, we found that forces exerted on neuroectoderm cells primarily impact on their movements and less on cell shape changes, suggesting a high degree of tissue fluidity, which has recently been identified as a key mechanical property in rearranging tissues[13,61–63].

Our combined experimental and theoretical results suggest mechanical transitions at different developmental times. Prior to internalisation, forces directed towards the posterior are likely to be generated by pulling (traction) forces of migrating mesendoderm front cells on the overlying neuroectoderm. Our model implies that only the front of the mesendoderm is actively migrating which agrees with our experimental observations. In contrast, long-lasting inter-tissue adhesion of mesendoderm rear cells enables opposing anterior-directed dragging of neuroectoderm cells. Asymmetric distribution of E-cadherin in the axial mesendoderm and differential adhesion to the adjacent neuroectoderm seem to play a key role in this process. The molecular underpinning of this process is currently unknown but might be regulated by distinct E-cadherin recruitment or trafficking pathways. Interestingly, Protocadherin-18a has been shown to be enriched specifically in the posterior axial mesendoderm where it regulates E-cadherin levels[64]. These observations raise the intriguing possibility that the mesendoderm collective is composed of different cell populations with distinct force-generating abilities, which is in

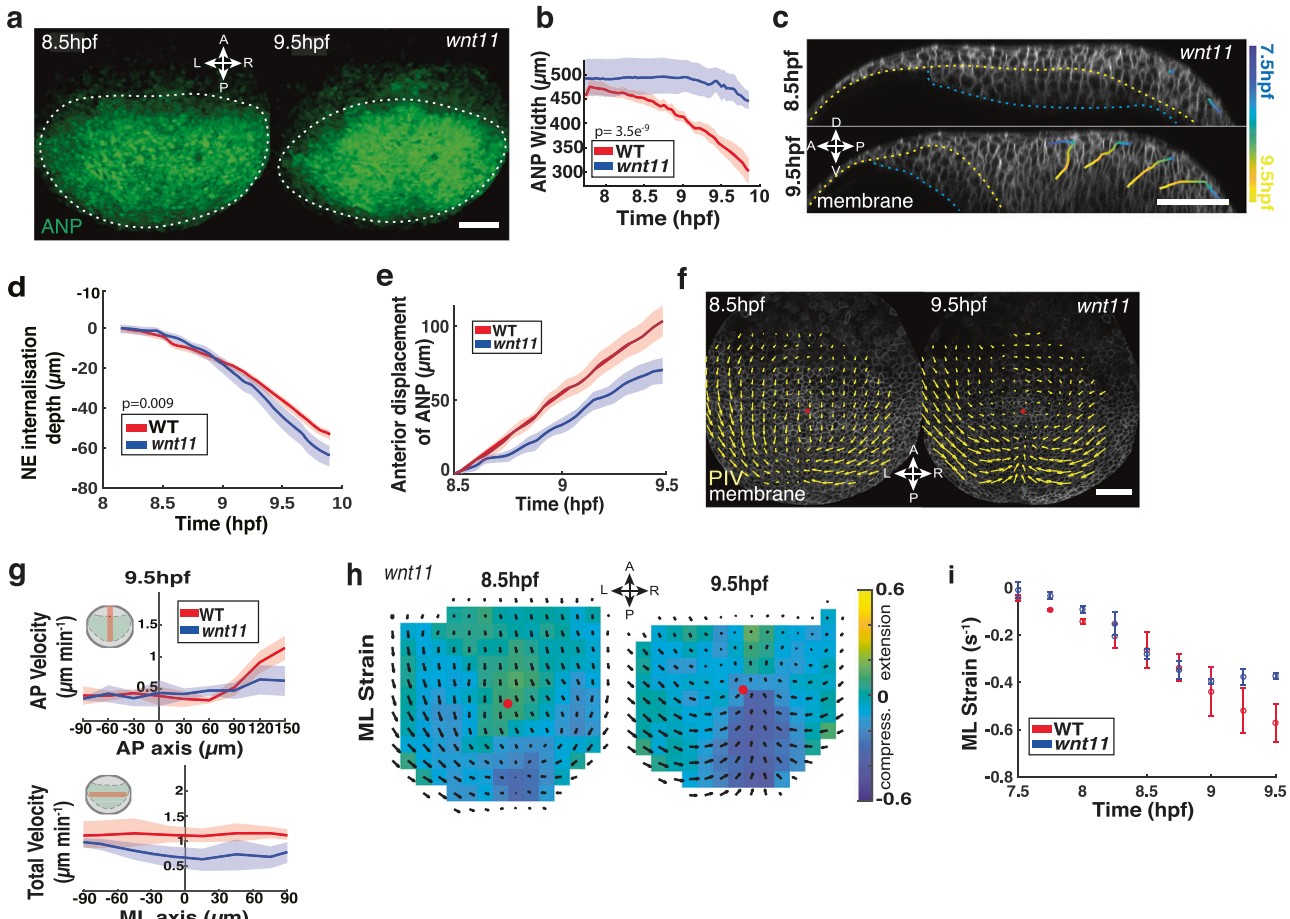

**Fig. 6 | Convergent extension is integral for anterior neural plate shape by supporting tissue extension. a** Confocal images of neuroectoderm (NE) cells in the anterior neural plate (ANP) of *Tg(otx2:Venus); wnt11* mutants during gastrulation (8.5–9.5 hpf). White dashes outline the ANP. Anterior (A), posterior (P), left (L), and right (R). **b** ANP width during gastrulation in WT (red) and *wnt11* (blue) mutants. Two-sided unpaired *t*-test (*p* = 3.5e⁻⁹). **c** Sagittal confocal images of representative cells tracked within the ANP in *wnt11* mutants through gastrulation. Dorsal (D), ventral (V), anterior (A) and posterior (P) indicated. Cell membranes (mRFP) grey LUT. Yellow dashes: NE/mesendoderm (ME) border. Blue dashes: ME/yolk interface. Tracks coloured blue (7.5 hpf) to yellow (9.5 hpf). **d** Internalisation depth of NE cells in WT and *wnt11* mutants during gastrulation (30 cells, 3 embryos). Two-sided unpaired *t*-test (*p* = 0.009). **e** Anterior displacement of internalising NE cells in WT and *wnt11* mutants (30 cells, 3 embryos). **f** Tissue flows of time-averaged velocities projected on confocal imaging of membrane-labelled

NE cells in the dorsal-most layer of the ANP in *wnt11* mutants at 8.5 hpf and 9.5 hpf. Red dot indicates intersection of AP axis with anterior edge of ANP tissue. **g** Cell velocities at 9.5 hpf in AP direction along dorsal midline, total velocities along mediolateral (ML) axis of WT and *wnt11* mutants. AP plot, X-axis: 0, anterior edge of the ANP; negative, anterior and positive posterior. Y-axis: negative posterior, positive anterior-directed flows. Total plot, X- axis: 0, dorsal midline; negative left, positive right. **h** Time-averaged ANP strain rates along the ML axis of *wnt11* mutants. Strain rate minimum green, maximum stretching yellow, maximum compression blue. Black arrows: time-averaged tissue flows. Red dot: AP axis/ANP leading edge intersection. **i** Maximum absolute strain rates along the ML axis of WT and *wnt11* mutants during gastrulation. Negative values represent compression. All data are presented as the mean ± SEM of 3 embryos, unless otherwise stated. Scale bars: 100 μm. Source data are provided as a Source Data file.

agreement with recent reports in zebrafish showing that front and rear cells of the migrating mesendoderm or the lateral line primordium, have specialised functions[65,66]. We propose that observed differential behaviour of mesendoderm cells can lead to interacting opposing flows in the ANP resulting in local mechanical instabilities that initiate internalisation and multilayer tissue folding. Out of plane forces (represented by a sink in our model) drive convergence in the ANP, which we infer as stresses generated by continuous internalisation of cells. This is accompanied by anterior-directed forces, which are likely to be produced by mesendoderm migration and pushing forces due to AP axis extension of posterior tissues, such as the notochord undergoing CE. This is in line with observations in the *Xenopus* neural tube where cell intercalation driven CE occurs primarily in the posterior region and pushes the anterior neural plate forward[14].

Notably, the molecular and cellular basis of the mechanism that we describe here, involving E-cadherin-mediated tissue coupling, differs from the previously reported mechanisms regulating similar

processes of neural plate morphogenesis during zebrafish neurulation. Convergence during neurulation is mainly driven by polarised cell intercalations and requires Planar Cell Polarity signalling and extracellular matrix between the neural plate and the adjacent mesendoderm[7,67,68]. Further, internalisation of cells in the zebrafish hindbrain during neurulation was recently shown to be dependent on intrinsic N-cadherin and myosin 2-dependent activities leading to cell surface constriction and internalisation along the dorsal surface[37]. By contrast, we found no clear indications for a contribution of N-cadherin or myosin 2 mediated contractility in the neuroectoderm to internalisation during gastrulation. This suggests that internalisation mechanisms in the neural plate may depend on developmental timing and the neighbouring microenvironment including the establishment of an extracellular matrix and apical-basal cell polarity in the neuroectoderm, which are not apparent during gastrulation.

Recent studies showed that diverse specialised cellular mechanisms control tissue folding in various organisms[69–74]. Folding is best

understood in polarised epithelial monolayers where tissue-intrinsic stresses lead to out of plane bending of a sheet which is often associated with cell internalisation or tissue invagination events in developing embryos[2,18]. Our findings provide insights into an alternative mechanism of (multilayer) tissue folding by a multi-tiered model by which differential inter-tissue adhesion regulates internalisation. In particular, the multiple, asymmetric stacked folds that we have observed here and that are parallel to the single layer of cells from which they form, are rather distinct from the usually symmetrical folds in simple epithelia. As noted above, our findings also differ from previously described mechanisms driving neural plate folding during neural tube formation in vertebrates. Neural tube morphogenesis is known to be regulated by actomyosin driven apical cell constriction, polarised cell intercalations of converging cells leading to axis extension and the contribution of apoptotic cells, which together generate intrinsic forces that drive formation of hinge points and bending of the neural plate[21,23,75–78]. However, recent work from mammalian spinal neural tube folding indicated that hinge points emerge passively in response to external forces[79]. Further mechanical in silico models will be needed to understand to what extent this hinge-point view applies to the internalisation events described here. More generally, mechanical descriptions of the complex tissue folding event that may arise in tissues in which multiple cell layers move relative to each other, as described here, are still largely missing.

Our work also suggests an important role for tissue shape and geometry in forebrain development. Various zebrafish mutants with brain and eye abnormalities are associated with a misshaped anterior neural plate at the end of gastrulation such as evident in mesoderm-less (Nodal/MZoep) or CE mutant embryos[32]. We speculate that morphological defects apparent in neurulation may be associated with earlier failures in proper neural plate shaping during gastrulation. The ANP gets subdivided into distinct domains which is regulated by signalling centres mostly located at domain and tissue boundaries[24]. Hence, the structure and geometry of the ANP tissue may control cell fate specification and positioning of domains during early tissue patterning. This idea is supported by recent studies showing that tissue geometry is crucial for the formation and patterning of reconstituted tissues and organs[80,81], and in particular during neural tube formation[58]. Future work will be needed to understand the functional role of shape and geometry in the emerging forebrain and how they feed back into mechanism regulating cell fate specification for early tissue patterning.

# Methods

## Fish lines and maintenance
Wild type (WT) AB and TU zebrafish lines were provided by the Biological Services Facility, University of Warwick. Transgenic lines: Tg(otx2:Venus)[33], Tg(actb1:lifeact-GFP)[82], Tg(gsc:EGFP-CAAX)[32], Tg(1.8gsc:GFP)[83]. Transgenic mutant homozygous line Tg(otx2:Venus); wnt11 was generated in-house by crossing Tg(otx2:Venus) with wnt11 f2 mutants [also called wnt11/slb (silberblick)][51]. To generate the Tg(gsc:Cdh1-EGFP) line, mouse E-cadherin-GFP-GI-polyA[41] was placed downstream of the goosecoid (gsc) promoter in the Tol2-basic vector[32,84]. Zebrafish were maintained as described previously[85] and embryos were raised at 28–31 °C in E3 buffer. Embryos were selected for imaging using established morphological criteria[86]. Adult zebrafish were maintained in the University of Warwick aquatics facility in compliance with the University of Warwick animal welfare and ethical review board (AWERB) and the UK home office animal welfare regulations, covered by the UK Home office licenses PEL 30/2308 and X59628BFC. The use of transgenic lines received ethical approval.

## Microinjections of capped mRNA and morpholino antisense oligonucleotides
Capped mRNA used for injection was synthesised using SP6 mMessage mMachineKit (Thermo Fisher Scientific Inc., MA, USA). For ubiquitous mRNA expression, 100 pg of mRFP mRNA, 100 pg of Lefty1 mRNA[87], 75–100 pg of h2afva-tagBFP mRNA[88], and 400 pg of dnRAC1 mRNA[45] were injected into one cell stage embryos. Morpholino oligonucleotides (MOs) were designed and synthesized by Gene Tools (Gene Tools LLC, OR, USA) and injected into one-cell stage embryos. To down regulate E-cadherin or N-cadherin, 3–4 ng cadherin 1[32] or 3 ng cadherin 2[89] MO was injected into one-cell-stage embryos, respectively. To generate mesoderm (prechordal plate) progenitors, 100 pg squint mRNA and 2 ng sox32/casanova MO[90] were injected into one cell stage embryos.

## Inhibition of Sonic hedgehog and nodal signalling
Transgenic Tg(otx2:mVenus) zebrafish embryos were microinjected with h2afva-tagBFP and mRFP mRNA, as previously described, and incubated in E3 media until shield stage. Embryos were manually dechorionated and transferred into E3 medium containing either 10 µM SHH inhibitor (KAAD-cyclopamine, MilliporeSigma™, MA, USA) or 50 µM Nodal inhibitor (SB-505124, AdooQ® Bioscience, CA, USA) and incubated for 1 h prior to imaging. Control embryos received identical treatment and were incubated in an equivalent concentration of DMSO. Embryos were scored at 24 hpf for presence of the inhibited phenotype to confirm inhibitor activity.

## Sample preparation for live cell imaging
Dechorionated embryos were mounted in 0.8% low-melting-point (LMP) agarose (Thermo Fisher Scientific Inc., MA, USA) into agarose moulds inside a glass fluorodish (World Precision Instruments, FL, USA) and covered with E3 medium with the animal pole of the embryo facing towards the objective. Wild type embryos were checked for normal development and mutant embryos for respective phenotype after imaging (12–24 h post fertilisation).

## Confocal imaging
In vivo live imaging and fixed whole mount imaging of embryos were performed using either a ZEISS 880 confocal microscope using a ZEISS 25x oil objective (NA = 0.8, ZEISS, BW, Germany) or an Olympus FLUOVIEW FV3000 confocal microscope (Olympus, Tokyo, Japan) and 20x UPLFLN objective lens (NA = 0.5) or 30X UPLSAPO objective lens (NA = 1.05). The temperature during imaging was kept at 29.5 °C using a temperature chamber. Embryos were mounted at 7.5 h post fertilisation (hpf) and imaged through to the end of gastrulation (10 hpf). BFP, mVenus/eGFP, and RFP were imaged using an excitation wavelength of 405 nm, 488 nm, and 564 nm respectively. Z-stacks were taken at a spacing of 1.2–1.25 µm with a XY resolution of 0.78–0.8 µm per pixel. Images were taken every 180–294 s.

## Multiphoton imaging
For in vivo live imaging, embryos were imaged using an Olympus FVMPE-RS inverted multiphoton microscope using a 30x silicon objective (NA = 1.05, Olympus Life Science, MA, USA). Proteins were imaged at the following wavelengths: mRFP – 1050 nm; Venus and GFP – 925 nm; BFP – 780 nm, using a Spectra-Physics InSight X3 and MaiTai HP DS-OL laser. Power was adjusted dependent on tissue depth and Z-spacing was set to 1.2 µm and a resolution of 0.53 µm per pixel. Images were taken every 180–220 s. Embryos were imaged at 29.5 °C.

## Whole-mount immunohistochemistry and antibodies
For whole-mount immunohistochemistry, embryos were fixed overnight at 70% (7.5 hpf), 80% (8.25 hpf), 90% (9 hpf) and 100% (10 hpf) in 4% paraformaldehyde in 1x PBS. After fixation they were washed 6 times in PBS with 0.1% Triton-X in 1x TBS (TBST) then permeabilised with 0.5% Triton-X in 1xTBS. Embryos were subsequently blocked in 0.1% Triton-X and 5% goat serum in 1xTBS. Phosphorylated myosin 2 was detected using a primary rabbit anti-phospho-myosin light chain 2 antibody (Cell Signaling Technology, MA, USA, #3671; 1/200 dilution).

Endogenous E-cadherin was detected using a primary mouse anti-E-cadherin antibody (BD Transduction Laboratories™, NJ, USA, catalogue number 610182; 1/100 dilution). Incubation with primary antibodies was performed overnight at 4 °C in TBS-T containing 0.1% Triton-X and 5% goat serum in 1xTBS at 4 °C. Embryos were subsequently washed with TBS-T six times for 10 min each and incubated overnight at 4 °C with either secondary antibody (Alexa 488-conjugated goat anti-rabbit, Thermo Fisher Scientific, MA, USA, A-11008; 1/5000 dilution) and rhodamine-phalloidin for F-actin staining (Thermo Fisher Scientific, MA, USA, R415; 1/200 dilution) for p-myosin, or goat anti-mouse Alexa Fluor™ 568 (Invitrogen™, MA, USA, catalogue number A-11004; 1/1000 dilution) and anti-F-actin conjugate Alexa Fluor™ 594 Phalloidin (Invitrogen™, MA, USA, catalogue number A-12381; 1/400 dilution) for E-cadherin. Embryos were washed three times for 5 min with TBST and nuclei were stained with either DAPI nuclei acid stain (Thermo Fisher Scientific, MA, USA, D1306) for p-myosin or Hoechst 33342 (Thermo Fisher Scientific catalogue number 62249) for E-cadherin for 30 min before a final round of three times 5 min washes in TBS-T.

### Transplantation assays

For cell transplantation experiments, donor and host embryos were kept in Danieau's solution (58 mM NaCl, 0.7 mM KCl, 0.4 mM MgSO$_4$, 0.6 mM Ca(NO$_3$)$_2$, 5 mM HEPES pH 7.6) after dechorionation. To produce mesendoderm (prechordal plate) progenitors, *Tg(gsc:GFP-CAAX)* donor embryos injected at one-cell stage with 100 pg of *squint* mRNA and 2 ng morpholino against *sox32/Casanova*[90]. Additional mRNAs or morpholinos (MO) were injected depending on the transplantation performed: for the DNRac1 phenotype, donors were also injected with 400 pg of *dnRAC1* mRNA[45] and 50 pg of *H2A-BFP* mRNA; for downregulation of cdh1, 2 ng of MO against *cdh1*[32]; host *Tg(1.8gsc:GFP)* or *Tg(gsc:EGFP-CAAX)* embryos were injected at one-cell stage with 50 pg of *mRFP* mRNA. Groups of mesoderm-induced cells (100–200 cells) were then removed from the animal pole of donor embryos at sphere stage using a glass transplantation needle (20 μm diameter) and transplanted below neuroectoderm cells ahead or into the rear of the mesendoderm collective in host embryos between 7 hpf and 8 hpf (before ANP internalisation). A membrane (*mRFP*) or nuclei (*H2A-mCherry* or *H2A-BFP*) marker was used to distinguish host and donor cells. Transplanted embryos were mounted for imaging as described above.

### Image processing

**Rotation and origin.** The otx2:Venus channel was used to identify the angle of rotation about the *z*-axis and identify the (*x,y*)-coordinates of the new origin, with the aim of aligning the front of the otx2:Venus region with the *x*-axis, placing the new origin at the centre of the anterior edge (Supplementary Fig. 1d–f). The time point corresponding to tissue internalisation was selected to compute these values as it represents the time when the anterior edge of the tissue is flat. The mean intensity *z*-projection of the channel was taken, and Otsu thresholding applied. The contour around the largest connected component of this thresholded image was taken to be the boundary of the otx2:Venus region. Lines between all points of the contour with length between 20 and 200 pixels were drawn to identify the modal unsigned direction, represented as the angle from the positive *x*-axis between 0 and π radians, using a bin width of π/10. All lines with an angle not in the modal or adjacent histogram bins were discarded, and the remaining lines were used to group points on the contour by connectivity. The largest set of nodes connected by these lines was taken as the points along the front (Supplementary Fig. 1e, red line). Points along the front were sorted by position along the contour, starting at the end of the front that preserves the handedness of the axes in the following step. The average signed direction angle of all remaining lines connecting points on the front to later points in the

sorted sequence was taken to be the new positive *x*-direction, *v*, and the new positive *y*-direction, *u*, was taken as perpendicular to this direction, pointing away from the otx2:Venus region the front. The origin in these new coordinates was given by the centre of mass of the otx2:Venus region in the direction *v*, and the centre of the points on the front in the direction *u*.

**Surface projection.** The whole embryo was segmented by applying Gaussian smoothing (*s* = 3 pixels) to the membrane channel followed by adaptive Phansalkar thresholding (radius = 30 voxels)[91] and selecting the largest connected component of the binary output as the segmented embryo. Points on the outer surface (Supplementary Fig. 1g) of the embryo were identified from the voxels with maximal *z*-component for each (*x,y*)-coordinate covered by the segmented embryo. For images with the outer surface facing *z* = 0, the minimal *z*-component was used. This orientation was detected by first computing the mean distance to the origin, defined above, of all pixels in the segmented embryo in each *z*-slice and taking the median of these values in the top and bottom halves of the *z*-stack. The half with the smaller median was taken to contain the outer surface, since it represents the narrower portion of the embryo in the imaged volume. The *z*-coordinate of the new origin was taken to be the *z*-coordinate of the surface point nearest the origin in the (*x,y*)-plane identified in the previous section at the same time point. All surface points were translated to place the new origin at (0,0,0). Each translated surface point, *p* = (*x,y,z*), was mapped to the point *q* = *r(p)*(*x,y*), with scale factor *r(p)* such that *q* and *p* have the same length. An inverse mapping from integer (*x,y*)-coordinates onto the surface was found by linear interpolation of *z* from projected surface points followed by Gaussian smoothing of *z*-values (*s* = 20 pixels), with the mapping completed by rescaling the (*x,y*)-coordinates of the surface points to maintain lengths as in the forward mapping. This yielded a smooth surface and a corresponding planar mapping to a 2D image. Volumetric mapping was achieved by translating surface points along the surface normal in steps equal to the original *z*-resolution (Supplementary Fig. 1g), yielding a stack of projected 2D images with fluorescence values mapped using nearest neighbour interpolation (Supplementary Fig. 1h, i). This mapping allowed standard image analysis methods to be applied to individual slices without having to contend with multiple layers of the embryo being present within a single slice.

### Image analysis and tissue flows

**Tissue shape analysis.** The shape of the tissue was calculated using the surface projection of the otx2:Venus signal as described above. The projection was then analysed by segmenting the maximum projection using a watershed algorithm in MATLAB (MATLAB version 9.5 R2018b/9.13 R2022b). Area of the tissue was calculated using this segmentation. The extent of the left-right axis was calculated by averaging the width of the tissue at five points along the anterior-posterior (AP) axis. All parameters were averaged over different embryos.

**2D and 3D cell segmentation and analysis of cell shapes.** Projected cells were segmented in 2D by first applying Gaussian smoothing (*s* = 3), followed by watershed segmentation of the mRFP channel. The segmented cell shapes were used to measure surface area, orientation and maximum and minimum axis using MATLAB's regionprops function. Values were refactored to account for changes induced by the mapping by taking locations of the extremities of the major and minor axes of the ellipse and projecting these back into 'original' coordinates. For calculations of the orientation and dynamics of 'domains' of cells, boxes with sides of 30 μm (-15 cells) were used and the average shape of cells within these domains was calculated over a period of 20 min. The centre of the domain was centred on an individual cell that was manually tracked from the anterior edge of the ANP. To investigate the dynamics of cells as they internalise, cells were manually backtracked

to mid-gastrulation. The apical and basal area of the cell was then measured using ImageJ (v1.53t, Wayne Rasband and contributors, National Institutes of Health, USA, Java version 1.8.0_322 64-bit)[92]. The dorso-ventral extent of cells was calculated by the distance between these sections, distances were averaged between multiple cells and multiple embryos.

Manually tracked cells were segmented in 3D (25 cells, 3 embryos). Matlab's 3D watershed segmentation function was applied after preprocessing (3D CLAHE, window size = 10 followed by Gaussian smoothing, σ = 2). The line marking the length of the cell in 2D was used to select labels. Voxels that the line passes through and neighbouring voxels were marked as *line voxels*. Labels intersecting at least 30% of the line voxels were taken to be part of the cell. Selected labels were dilated by 1 voxel in each direction, yielding the final segmentation. The segmentations were manually verified against the original images in ImageJ, and all over- and under-segmented frames discarded (success rate: 76%). Cell lengths were computed by fitting an ellipsoid to the segmentation mask in Matlab, and taking the axis most closely aligned with the 2D length measurement.

2D Segmentation of 135 cells in the ANP region of 3 embryos were compared to manually annotated ground truth to confirm accuracy. Cells were taken from 5 frames distributed over the length of each recording. In each frame, 3 cells were randomly selected from each of 3 distinct regions of the ANP, given by the areas 75 μm from the dorsal midline, 75 μm from the lateral edges of the ANP, and the areas between these two regions. Cells considered successfully segmented (intersection over union >0.6) made up 76% of the sample. Area, orientation, and length measurements all had strong correlation between manual and automatic segmentation (0.76, 0.79, and 0.47, respectively). Mean area and length differences relative to cell size were −0.04 and −0.02, respectively, while mean angle difference was 4.8 degrees, suggesting that measurement errors have low bias. No correlation between time or position of cells and error was observed.

**Analysis of cell depth during internalization.** To determine the depth of individual neuroectoderm cells in the ANP over the period of gastrulation, cells that were initially within the dorsal cell layer along the dorsal midline of the embryo were manually tracked within multiple embryos. The minimum distance to the surface of the embryo was calculated.

**Quantifying tissue flows.** Projected images were analysed using the MATLAB application PIVlab. 30 μm² boxes were created spanning from the origin along the lateral and AP axes. Velocities of cells within these boxes were then averaged over 15 min to calculate the average velocity of domains within the tissue.

**Measuring heterotypic cell-cell interactions.** To calculate the contact time between mesendoderm and neuroectoderm cells in the ANP, mesendoderm cells were manually tracked. The time period for each new neuroectodermal cell contact was then averaged over the period of imaging to give a mean time of cell contact. This was completed for a point at the front (anterior), the middle and the rear (posterior, adjacent to the notochord) of the mesendoderm. Contact times were then averaged over different embryos.

**Analysis of cell-level otx2:Venus expression.** Confocal time series images of *Tg(otx2:mVenus)* embryos were imported into Imaris (v9.9.1, Oxford Instruments, Oxfordshire, UK). 15 ANP and 15 non-ANP cells were identified per embryo at the end of gastrulation and manually tracked backwards to the start of the image set. Otx2:Venus expression levels per frame were extracted and imported into GraphPad Prism (version 9.1.0 for Windows, GraphPad Software, San Diego, California USA, www.graphpad.com). Expression levels were normalised per embryo, and data are presented as the mean ± SEM of the 3 embryos.

**Quantification of E-cadherin, lifeact-GFP and p-Myosin 2 levels in live and fixed embryos.** Measurements were performed in ImageJ/Fiji using sagittal sections from live imaging of gsc:cdh1-EGFP, gsc:EGFP control or lifeact:EGFP at -8 hpf (before internalisation) in the embryo through the reslice function. A freehand line was drawn in the mesendoderm close to the neuroectoderm interface, starting from the front to the rear of the collective, covering around 250 μm of tissue. Plot profiles of signal intensity were generated for three different sections per experimental conditions, normalised and averaged over all sections. For quantification of lifeact:EGFP, a segmented line was use to trace the perimeter of 10 cells in the front and 10 cells in the back before internalisation. Tracing was started at the side of the cell furthest from the direction of movement. Plot profiles of signal intensity were generated for three different embryos, normalised and averaged.

Line scans were also performed across the ANP perpendicular to the midline to quantify cell-level p-myosin expression in fixed embryos. The plot function was used to quantify the fluorescence by position in the p-myosin and nuclei channels. Data were imported into GraphPad Prism and 2nd order smoothing applied using 10 nearest neighbours. Data are presented as the raw and smoothed data from 3 embryos.

To detect endogenous E-cadherin levels, z-stacks of cdh1-stained embryos were imported into ImageJ and the line scan tool was used to measure the average grey value at the cell-cell contacts between mesendoderm cells and the overlying neuroectoderm. 10 cell-cell contacts were measured for cells at the front of the collective (within 50 μm of the leading edge) and 10 cell-cell contacts were measured for cells towards the rear of the collective (around 150 μm from the leading edge). Grey values were normalised for each embryo ($n = 3$) and data are presented as individual cdh1 expression levels with the mean ± SEM. An unpaired *t*-test was conducted in GraphPad Prism to compare the front and rear expression levels (*p*-value < 0.0001).

**Quantification of cell protrusions.** Cell protrusions of donor WT control and DN Rac1 cells in multiple embryos were manually counted throughout the length of the movie (7.5–9.5hpf). Cell protrusions in *Tg(actb1:lifeact-GFP)* embryos were manually counted in 20 cells of the front (within 50 μm of the leading edge) and 20 cells of the rear (around 200 μm from the leading edge) of the mesendoderm tissue at the interface with the neuroectoderm before internalization, over a period of ~30 min, and expressed as number of protrusions observed in that region per minute.

**Quantification of cell velocities.** Mesendoderm front cells (within 50 μm of the leading edge) and cells in the rear (around 150 μm from the leading edge) of the collective were manually tracked in ImageJ using the TrackMate plugin. XYZ coordinate pairs were extracted for each time point to calculate the instantaneous speed of each cell at each time point. Data were imported into GraphPad Prism and 2nd order smoothing was applied using 7 nearest neighbours. Data are presented as the mean ± SEM and smoothed mean of 5 front and 5 rear cells from each of 3 embryos.

## Calculation of tissue strain rates
Strain rates in the neuroectoderm were calculated from the tissue flows using spatial derivatives of velocities in neighbouring domains (~40 μm²)[38]. The strain rates in the anterior-posterior (AP) and left-right (LR) directions are:

$$\varepsilon_{AV} = \frac{\partial v}{\partial y}, \varepsilon_{LR} = \frac{\partial u}{\partial x},$$

respectively, where $x$ and $y$ are the coordinates along the LR and AP axes, and $u$, $v$ are the velocities in these directions. These strain rates

determine the stretch (positive strain rate) or compression (negative strain rate) in these directions.

## Statistical and data analysis

Data was stored in mat files and analysis was performed using MATLAB. All data plots were generated in MATLAB using built-in functions. Box plots generated using the boxplot function; central line indicates the median, and the bottom and top edges of the box indicate the 25th and 75th percentiles, respectively. The whiskers extend to the most extreme data points not considered outliers, and the outliers are plotted individually using the + marker symbol. A one-way nonparametric Kruskal–Wallis test was used when comparing whether different groups came from the same distribution and implemented using the MATLAB kruskal–wallis function. Other comparisons between two groups were performed by using unpaired $t$-test or two-sided Wilcoxon rank-sum test, implemented using the MATLAB ranksum function.

## Theoretical model

Details of the theoretical model can be found in the Supplemental Note.

## Reporting summary

Further information on research design is available in the Nature Portfolio Reporting Summary linked to this article.

## Data availability

Results and data files from the modelling is provided at https://zenodo.org/records/15458567. Source data are provided with this paper.

## Code availability

The code generated in this study for image processing and surface projection is provided at https://doi.org/10.5281/zenodo.8171784 under the MIT software license. A supporting text describing the modelling and fitting procedures is available in the supplementary note. The codes used for data analysis and simulations are available at https://zenodo.org/records/15458516.

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

## Acknowledgements

We are thankful to all members of our labs for continuous support and comments on the manuscript. We thank Yanlan Mao and Timothy Saunders for helpful suggestions and comments. We are thankful to Michael Brand and Carl-Philipp Heisenberg for providing fish lines for the work. We are grateful to the Warwick BSU aquatics facility for zebrafish care and CAMDU (Computing and Advanced Microscopy Unit) for their support and assistance during this project. This work was supported by the Medical Research Council (MRC) Doctoral Training Partnership (MR/N014294) to A.I., the Biotechnology and Biological Sciences Research Council (BBSRC) MIB Doctoral Training Partnership (BB/T00746X/1) to E.S., an EPSRC grant award (EP/V062522/1) to T.B., the Max Planck Society to P.A.H. and a BBSRC grant award (BB/T016492/1) and the Warwick Quantitative Biomedicine Programme funded by the Wellcome Trust Institutional Strategic Support Fund (ISSF) to M.S.

## Author contributions

A.I. and M.S. conceived the project and designed the research. A.I., E.S. and B.L.E. performed the experiments and analysed the data. J.E.L. and T.B. performed computational image processing and analysis. P.A.H. performed the numerical simulations and modelling. M.T. provided reagents and conceptual input. A.I. and M.S. wrote the original manuscript. All authors reviewed and edited the manuscript.

## Competing interests

The authors declare no competing interests.
