## [Transparent Peer Review file · Nature Communications]

A multi-tiered mechanical mechanism shapes the early neural plate

Corresponding Author: Dr Michael Smutny

Version 0:

Reviewer comments:

Reviewer #1

(Remarks to the Author)

The manuscript represents a large body of work, including the generation of some useful transgenic lines. It is an elegant combination of flow quantification from wt data, in silico modelling and genetic perturbations, generating mechanistic conclusions about how the complex process of anterior neural plate morphogenesis has arisen from cell-scale to tissue level. It is well-worthy of publication in nature communications. I have some points to clarify below, which I believe are all minor.

Introduction

- The introduction is clearly written but a more detailed introduction into the use of zebrafish anterior neural plate as a model and how it relates to mammalian neural plate morphogenesis would be helpful. What kind of morphogenetic movements occur (e.g. tissue folding and anterior elongation) and are these similar to those that occur in the amniote neural plate. How does this relate to the later structure (i.e. neural tube).

Results

- For all figures that are relevant, please include how many cells were analysed in the legend.
- Although there might be space constraints, it would be helpful if wt panels could be inserted alongside perturbed panels (e.g. of flow topologies) to enable easier comparison.
- Throughout the manuscript, 'strain rates' are discussed in the text but the figure labels say 'strain'. These are two different measurements. Can this please be clarified.
- Figures would benefit from more extensive labelling, especially for axes such as A-P and D-V. Marks to clearly indicate points of interest (e.g. flow collision points) would also be helpful
- Fig 1K&L. I can't make out the details of the blue lines in I.
- Supplementary figure 2a: axis labels would be helpful.
- Line 167. I suggest that referencing the Araya paper (ref 45) detailing pMyosin and Myl12.1-GFP upregulation at the midline internalisation site in the hindbrain would be pertinent at this point.
- Supplementary video 2, figure 2a, supp fig 2b: Annotation would make it clearer. Also, in the video legend ventral should be down, not 'right'.
- Figure 2C. I am confused by these strain rate maps. Some of this may be remedied by fuller explanation in the figure legend and potentially combining supplementary figure 2d with figure 2c. It looks like the strain-rate heat map is overlying the velocity vector maps, similar to those shown in figure 1f. In that case, how have the AP and ML strain rates been separated on the PIV plot? I think the vectors should be corresponding to anisotropic strain rate and not velocity. If the tissue is internalising at this point, would this also give a similar strain rate map, without compression having occurred? I think the data and conclusions are fine – it just needs clearer presentation.
- I am not able to assess the supplemental note describing the mathematical model, since this is outside my expertise, but the output shown in figure 3 is interesting and makes sense.
- I would have found it easier to follow if the "Internalisation depends on E-cadherin-mediated tissue interaction" section was after the "Differential behaviours of mesendoderm cells initiate ANP tissue folding and cell internalisation" section.

Discussion

- One thing that I did find a bit confusing throughout the manuscript was the overall message of how the flows and compression points linked to the folding and internalisation events. All the information is there in the manuscript, but it would

benefit from a summary figure and/or conclusions section. Figures 1m and 2k are helpful. Perhaps some kind of combination of the two, showing the morphogenetic movements that occur in the ANP and how the flows link to this.

- I suggest that the manuscript needs a clearer explanation of what is meant by ANP folding. Is this really tissue folding (which is more usually associated with an already polarised epithelium) or is it more akin to 'cell internalisation', as described in reference 45? This could be incorporated into the suggested summary figure in the point above.
- Another thing that is currently missing is a fuller explanation of how ANP morphogenesis and the results of this manuscript fit within:
 - a) Zebrafish neurulation
 - how is this early neural plate shape important for later neurulation? E.g. how does this early folding event relate to the later folding events described in the same tissue (e.g. reference 71 Brewster lab)?
 - How does ANP morphogenesis relate to hindbrain neural plate morphogenesis (e.g. reference 45 Araya paper)? Why does it occur via a different mechanism (i.e. not via NMYII)?
 - b) Vertebrate neurulation
 - How do the morphogenetic events described here relate to amniote neural plate morphogenesis and later neurulation?
- I suggest that the final sentence of the discussion section is a bit out of place and could be replaced by a stronger concluding sentence.

Methods

- Can you indicate on figure legends which imaging modality was used?

Clare Buckley

Reviewer #2

(Remarks to the Author)

In their manuscript, Inman et al analyze the morphogenesis of the anterior neural plate in zebrafish embryos using quantitative imaging and genetics. The anterior neural plate rests on top of the mesendoderm and folds during neurulation. The authors use a transgenic line to label the anterior neural plate and also label the membranes of the cells. This allows them to look at tissue movement/flow. They then developed an image software pipeline and projected the anterior neural plate in a 2d-plane and extracted the movement and shape changes. Using an approach from Richard Adams' group, they then calculate the anterior-posterior and medial-lateral strain rates. Together, these approaches show that the anterior neural cells move towards the midline and are compressed along the midline as the tissue folds. This leads the authors to propose a model in which the mesendoderm (that is located underneath the anterior neural plate) pulls on the neural cells in the front and restrains the neural cells in the back. The authors go on to model this process and find that their idea of how the mesendoderm pulls on the anterior neural plate can generate the observed flow patterns *in silico*. To support their model molecularly they assess whether the anterior neural plate adheres differentially along the AP-axis (they look at cell displacement as a measure for this) and whether F-actin levels differ along the AP-axis. They find that the front of the mesendoderm slides/moves along the neural plate while the back does not and that there is more Lifeact-GFP in the front and more *gfp*-tagged *cdh1*. They then perform functional assays. They block mesendoderm formation and find that the neural plate does not fold. They then place mesendoderm cells expressing *dnRac1* in front of the anterior neural plate (these cells are assumed to be immotile) and find that mesendoderm cells do not move much and the neural plate does not fold. They also test the role of convergent-extension movements using *wnt11* mutant and find that this does not affect folding. Lastly, they reduce the levels of *cdh1* using morpholinos and find that adhesion between the neural plate and mesendoderm was reduced (judged again by cell-cell displacement) and folding was also affected. The strengths of this study are the nice quantitative image analyses (which are mostly well documented and convincing) and the weaknesses are the molecular experiments that do not provide much mechanistic insight into the forces underlying neural plate folding and are preliminary at the current stage. I therefore do not think that this manuscript merits publication in a high profile journal in its current form.

Major concerns.

1. The authors find that the mesendoderm interacts with the overlying neural plate differently in its front and back. One key insight would be to know the molecular nature of this difference – this is a tough question to answer but would definitely move this manuscript from a more descriptive to a more mechanistic study.
2. Image segmentation. From the images in the figures it is hard to judge how well the cell segmentation worked. They should quantify this. How many cells are correctly isolated? Also, it would be nice to know how the watershed segmentation was done. Did the authors use marker-controlled watershed segmentation?
3. *Cdh1* analysis. Expressing *cdh1* from the *gsc* promoter does not report on levels of *cdh1* (the *gsc:gfp* is not a good control since it is a small transgene at a different insertion site). There are knock-in lines for *cdh1* available (PMID: 30969163). These lines should be used to assess levels and localization of *cdh1*. Also, overall levels are not key here. Key is how much *cdh1* localizes at the interface of the mesendoderm and the neural plate. In fact it is surprising that overexpression of *cdh1* in the mesendoderm does not affect the embryo.
4. Lifeact analysis. Similar to *cdh1*, total levels of lifeact-*gfp* are meaningless. Those levels depend on the promoter (b-actin in this case) and not f-actin levels. The authors need to look at subcellular localization and ask whether there is more localized lifeact in the cells in the front and rear.

5. Transplantation. The authors need to do control transplantations. Wt into wt – does placing cells in front of the neural plate affect morphogenesis.? Also, the clones should be positively marked, ie RFP expression and not absence of RFP expression in an RFP-expressing host. Absence is impossible to see/score. Also, the authors need to confirm that dnRac1 expression affects actin dynamics and cell movements as they assume.

6. The model the authors put forward argues that the mesendodermal cells in the front migrate forward pulling on the overlying neural cells and pushing them backward. The rear mesendodermal cells – in contrast – are held back by the overlying rear neural cells. So, the mesendodermal cells in the front are pulled forward while the mesendodermal cells in the back are held back. Since the tissue does not snap in two, mesendodermal cells need to fill in or mesendodermal cells need to be stretched (increase in strain rate). Do the authors see such kind of cell behavior in the mesendoderm?

Minor

1. The authors use neural cell to mesendoderm cell neighbor distance as a measure of adhesion. This could be correct but it could also reflect different degrees of mobility so strictly speaking this is not a direct measure of adhesion.

Reviewer #3

(Remarks to the Author)

The manuscript by Inman et al. describes the cell and tissue shape changes that occur in the anterior neural plate of Zebrafish during gastrulation. The authors first examine changes in the shape of the anterior neural plate using the Tg(otx2::Venus) reporter. They then map tissue flows across the embryo using PIV, which identified planar rotational flows and convergence and extension movements. Modeling of neuroectoderm tissue flows predicted that mechanical transitions at distinct stages can lead to observed flows. The authors then performed experiments removing or preventing migration of mesendoderm to suggest a role for mesendoderm migration in neuroectoderm internalization. Finally, they showed that internalization dependend on E-Cadherin, but not convergence and extension or N-cadherin. Overall, the manuscript described the early shape change of the anterior neural plate, but I did not find the mechanistic conclusions about tissue shaping to be well supported by the data, as I describe in my point-by-point response.

Main Comments:

1) Lines 86-108: The authors conclude that the anterior neural plate is dramatically reshaped throughout gastrulation. An alternative model is that there are changes in gene expression for the otx2::Venus reporter. The tracking of flows with PIV does not address this point, because individual cells and their otx2 expression status are not tracked and the third dimension is not accounted for. The authors should comment on previous data or show how the status of otx2::Venus expression changes (or doesn't) during this reshaping process.

2) Lines 139-140: The authors say that the convergence of lateral cells results in cell internalization that 'resembles tissue folding'. However, the images in Fig. 1i look more like a thickening of the tissue rather than a fold (there is no indentation at the surface). This stage does not seem to approximate the tissue folding that occurs later in neurulation, which was shown to have similarities with primary neurulation in mammals (Werner et al. 2021). I disagree with the use of 'tissue folding' throughout this paper, that language is misleading and creates a false analogy to other tissue folding processes.

3) Fig 2 and Supplemental Figure 2: When the authors see cell stretching, how do they know they are seeing the whole cell in each time step? Would be stronger evidence to show 3D reconstruction of entire cell. Also, cell stretching seems to happen after the cell has internalized, which is inconsistent with the language in Lines 169-172.

4) Lines 342-344 and Fig.4: An alternative model to the mesendoderm providing a force-generating role is that the mesendoderm provides a biochemical signal and that the position of this signal is important for proper internalization. The brute force approach of removing or preventing mesendoderm movement do not rule out the signaling model or definitively shown the importance of force generation.

5) Supplemental Fig. 5 and Fig. 6: The authors say that heterotypic interactions between N- and E-cadherin underlie the interaction between neural plate and mesendoderm. Thus, if a force-based mechanism existed for mesendoderm to internalize neuroectoderm, it would require both N- and E-cadherin. The fact that cdh2 morphants did not affect internalization argues against the authors point. The cdh1 morphants likely affects many tissue types and doesn't necessarily argue for a direct connection between mesendoderm and neuroectoderm.

Minor Comments:

1) Supplemental Fig. 2b, c: The cells in the region look multilayered and there are no polarity markers being used, so what enables the authors to define apical vs. basal?

2) Line 164-167: If internalization is stepwise as shown in Fig. 1j, then why would the authors expect that myosin II would be regionally enriched? It could be enriched in single cells that undergo internalization.

3) Line 186: The authors argue that tissue compaction acts as a hinge point, but there is no bending at the surface of the tissue in the movies or figures. This language is imprecise and misleading, I suggest the authors reword it. It looks like tissue compaction is leading to tissue thickening, not hinge point formation.

4) Fig. 2j: The actin in the image doesn't look cortical. Has it been processed (smoothed) in some way that was not described in the figure legend?

5) The term multi-tiered is never explained in the manuscript.

Version 1:

Reviewer comments:

Reviewer #1

(Remarks to the Author)

The authors have addressed my comments. Regarding figure 1K&L, I meant that I couldn't see the blue line histograms due to the small size of the figure. This is a minor point, but the clarity of the figure could be improved here.

(Remarks on code availability)

Reviewer #2

(Remarks to the Author)

In their revised manuscript, Inman et al. addressed a few of my concerns but the main issues (lack of mechanistic insight and lack of convincing/well-controlled experiments) were not addressed. As said initially, the quantification part looks solid but the molecular aspect of the study is preliminary/not convincing and potentially flawed. I therefore do not support publication of this manuscript. See details below and comments for the initial submission.

1. Cdh1 analysis.

There are a few issues with the authors analysis/argumentation.

- Gsc:cdh1-gfp. The expression levels of cdh1-gfp depend on the promoter gsc and the insertion site of the transgene, so any differences in expression are reporting on the promoter activity in the different cells and the effect of the insertion site and not on any differences in cdh1 expression. As I said this is pretty much meaningless.
- Effect of insertion site on transgene expression. Insertion sites do not only affect levels of transgene expression but expression also varies between cells depending on the insertion site. This is an unfortunate fact that makes the quantification of the expression of small transgenes pretty useless.
- Cdh1 antibody staining. This is a much better approach but the authors should show a complete image of the front and rear from the same embryo. Antibody stainings vary a lot between embryos. About 5-fold within a batch of embryos stained together. So an image similar to Fig. 2g is needed. The Fig. S2i is split and does not help.
- Cdh1 knock-in lines. The timeline for revision set by the editors should allow for the import of lines. This is more of a comment for the editors than the authors. Since the manuscript was initially submitted 4 months ago, there was ample of time to receive the lines. So I do not think this is a valid excuse for not doing the experiment.
- The differences in Cdh1 levels seen in Fig. 2g are likely due to the gsc promoter activity at the given genomic insertion site and not due to differences in cdh1 promoter activity (since it is the gsc promoter, the knock-in would address this) or altered cdh1 localization within the cell (levels are fairly the same along the cell membranes in the antibody stainings). Differential turnover, as the authors point out, is possible but to make that claim they need to use the endogenous promoter to exclude differences in promoter activity.
- Minor. In my opinion the statement that "These findings also establish that the Tg(gsc:cdh1-EGFP) line that we generated is a valuable tool to visualise internal E-cadherin dynamics" is incorrect. I would not see why anybody would use this line over the knock-in lines. This line is creating a completely artificial scenario with cells mis-expressing cdh1. Mis-expressed cdh1 will not reflect the endogenous cdh1 and will likely also not resemble the endogenous protein dynamics and may cause unwanted phenotypes.

2. LifeAct analysis.

- The quantification in Fig. 2n shows that both the front and rear cells have enriched LifeAct-GFP in their fronts. So, this is not supporting the authors hypothesis...
- More importantly, I have seen many LifeAct-FP images and movies and one normally sees nicely the outline of the cell cortex and cellular protrusions. I cannot see this in Fig. 2l. I would have guessed this is an image of a cytosolic FP. So much better images are needed.
- The authors claim that there are protrusions and provide a quantification in Fig. 2m but the primary movies of protrusions are missing. Since none of the images shows protrusions this is needed.
- As stated before, the intensity profile of LifeAct-GFP reflects reporter activity differences between the front and rear, not F-actin polymerization differences. This requires subcellular quantifications in Figs. 2l, S2j, S2k.

3. Transplantation experiment (Fig. 3l-n).

- The authors decided not to address my concern that donor and host cells need to be labeled in different colors. In the cdh1 mosaic analysis, the donor cells are labeled with membrane-bound GFP (GFP-CaaX). The highlighted cells do not show a membrane labeling although they should. The host cells are labeled with H2A-mCherry. However, nuclear labeling in the host cells is not discernable and the red channel is also shown as a gray scale (as is the green channel so one does not know what is what). The host cells are also labeled with cytoplasmic GFP so all host cells should be evenly green (or white given that the authors chose a gray LUT for both channels). This is not evident in the images. So, based on these data, the conclusions are not supported. See my initial comments.
- Similarly, in the dnRac1 mosaic analysis, the donor and host cells are labeled with a memGFP and the host cells express RFP (that oddly seems to be membrane bound). As said before, the clones need to be labeled with a FP. Absence of

expression is not suitable for many obvious reasons – the main one is that no tg or injected mRNA is expressed in all cells so there will always be unlabeled host cells.

- Quantification of the absence/reduction of protrusions in dnRac1 cells in Fig. S4j. The data need to be shown. Since none of the images/movies show protrusions, this is essential.

(Remarks on code availability)

Reviewer #3

(Remarks to the Author)

This manuscript details a complex morphogenetic process in vertebrates, showing how interactions between multiple tissues sculpts the anterior neural plate. The authors did a nice job addressing my comments and I just have some minor comments regarding clarity of the manuscript.

Line 22: The use of the word 'novel' could be misinterpreted. This is novel for the zebrafish ANP, but adhesive mechanisms resulting in friction is a general mechanism of morphogenesis.

Line 218: typo: enablingt  enabling

Line 427: typo: contorl  control

Line 590, ref 58: I believe the authors have the wrong reference. The paper they cite is a Drosophila paper. I think the meant PMID: 34707290. If they are looking for Drosophila papers to make this point they could consider PMID: 28504247 and 36724258

Figures: In general it would be helpful to have embryonic axes specified throughout the figures, similar for what was done in Supp Fig. 5K

(Remarks on code availability)

Version 2:

Reviewer comments:

Reviewer #2

(Remarks to the Author)

Comments for Inman et al

Cdh1.

Cdh1 knock-in line. This comment was addressed to both the editors and the authors. The authors chose not to get the line, which is odd because this would be the perfect experiment. The Cdh1 AB staining is ok but the localization to the membranes in Fig. 2g or Fig. S3a is not visible – one cannot see the membranes. So, this claim is not supported. However, Fig. S3a clearly shows more Cdh1 in the ectoderm than the mesoderm. Whether this difference in Cdh1 levels demarcates the neuroectoderm-mesoderm border/tissue interface is impossible to say from Fig. S3a (the dotted line the authors draw is partly going through nuclei, which is likely not the border...). Again, the Cdh1 knock-in line would likely answers this...

Lifeact.

Convincing images of protrusions: The authors now provided ok images of the protrusions in Fig. 2i

Protrusion dynamics: The authors ignored the request to provide a movie that shows the protrusion dynamics and is the basis for their quantification in Fig. 2j.

Transplantation.

The authors did not address my comments regarding the transplants. Instead of copying my precious comments again, the authors might want to show Fig. 3n to a third person, ask them whether they can see cells with a nuclear label or a membrane label. I cannot. Alternatively, the editors might want to have a look at Fig. 3n and see if they can make out cells with a nuclear label or a membrane label. The concerns are also valid for the dnRac transplants.

To me, key aspects of the claims the authors are making are not supported by their data. So, I do not think these should be published. Maybe one way forward is to remove the data from the manuscript and show the data that are solid. And adjust the conclusions.

(Remarks on code availability)

Version 3:

Reviewer comments:

Reviewer #1

(Remarks to the Author)

1. Cdh1

I can confirm the author's original response to this point – it is currently very challenging to import zebrafish lines from abroad due to import restrictions and high cost. If I had been in the same position, I also would have been hesitant to take this route, since even after transport, it would take 4-6 months to grow the line to breeding stage, significantly delaying publication. The transport of adult fish is not feasible with the current regulations. Also, the Cdh1 knock in lines are not specific to the mesendoderm, which would make visualisation of this tissue problematic (other overlying tissues would also be labelled). While there may be some merit in reviewer 2's concerns regarding reporting promoter levels rather than endogenous levels of Cdh1, fusion transgenes like this are regularly used to report the location of proteins within specific populations of cells. In my view, the authors have included sufficient additional experiments to confirm their findings from their fusion transgenic line: Immunohistochemistry staining clearly indicates endogenous levels of protein. Since the authors are interested in relative levels of Cdh1 at the mesendoderm/neuroectoderm interface between the rear and front cells and quantify fluorescence levels within individual embryos, this is an appropriate analysis.

2. Lifeact

The authors supplied the video that reviewer 2 requested

3. Transplantation

Whilst the multiple colours within both host and donor cells make it slightly challenging to understand this data initially, the cell populations are clearly and distinctly labelled in supplementary figure 4n and supplementary figure 5h. This data is appropriate.

In my view, the authors have now addressed reviewer 2's comments and the manuscript should be published.

Clare Buckley

(Remarks on code availability)

Reviewer #2

(Remarks to the Author)

Comments for Inman et al.

1. Cdh1 analysis

As stated in the last round of comments, the Cdh1 AB staining is ok. The Cdh1 membrane signal in the revised figures 2 and S3 is still not very convincing but ok. One worry that the authors may want to think about is that the high Cdh1 signal in the NE and the low Cdh1 signal in the ME is mirrored by the high nuclear signal in the NE and the low nuclear signal in the ME, suggesting that they might be looking at a penetration problem rather than a difference in Cdh1 expression levels – deeper tissues might simply be less well stained.

2. Lifeact analysis

The movies shown in video 3 are of low quality. It is impossible to make out individual cell protrusions because the temporal resolution is very low and the signal-to-noise is also not great – pseudo-coloring the movies also makes it harder to judge. Nature Methods recently had a comment on how to present fluorescence images, including the advice to always show the grey scale image. Maybe the authors want to heed that advice. The fact that the cells enrich F-actin and form protrusions is supported by the movies so this is ok. However, the authors should label the movies to make it easier for the reader (state the units of the time stamp, provide a scale bar, label the left and right movie in the video itself and not only in the legend).

3. Chimeric analysis

a. Fig. S4n

The cytoplasmic signal for gsc:GFP (host) and the membrane signal for gsc:GFP-caax (donor) cannot be distinguished. There are no cells with GFP on the membrane. GFP looks cytoplasmic, with higher levels in some cells and lower levels in other cells. The membrane label of mRFP (donor) is detectable but the nuclear H2A-RFP label of the host cells is not. The nuclear H2A-BFP label of the donor cells is also not detectable, at best there is some slight enrichment of BFP in some of

the cells but there is also the same faint (though a bit weaker) signal in all of the cells. As stated twice before, this is insufficient to support the claims by the authors.

b. Fig. 5i and S5h

The labeling of the donor cells with H2A-BFP is acceptable in these panels. It looks like not all donor cells are labeled with H2A-BFP but a large fraction. Also, the membrane GFP labeling of the host cells is visible. So, this is ok and supports the authors conclusion. It is unclear whether all the data were generated using such labeling or not though.

c. labeling of donor cells

A “qualitative” labeling of donor cells is not sufficient for mosaic analysis nor is it “optional” as the authors propose. Any worm, fly, mouse or fish geneticist would strongly object. One needs to know the genotype of the cells analyzed to correlate genotype with phenotype. It is a bit surprising that this needs to be discussed.

In summary, the conclusions pertaining to the dnRac experiments are now supported by data. However, the conclusion about the tissue/cell-specific role of *cdh1* is not supported by the data presented and should either be improved or – as suggested before – removed from the paper if the editors agree.

(Remarks on code availability)

Reviewer's comments are in **black**, author's responses are in **blue**.

Reviewer #1 (Remarks to the Author):

The manuscript represents a large body of work, including the generation of some useful transgenic lines. It is an elegant combination of flow quantification from wt data, in silico modelling and genetic perturbations, generating mechanistic conclusions about how the complex process of anterior neural plate morphogenesis has arisen from cell-scale to tissue level. It is well-worthy of publication in nature communications. I have some points to clarify below, which I believe are all minor.

Introduction

- The introduction is clearly written but a more detailed introduction into the use of zebrafish anterior neural plate as a model and how it relates to mammalian neural plate morphogenesis would be helpful. What kind of morphogenetic movements occur (e.g. tissue folding and anterior elongation) and are these similar to those that occur in the amniote neural plate. How does this relate to the later structure (i.e. neural tube).

We have added more information about neural plate development in the introduction as suggested. We have also addressed some of these questions extensively in the Discussion, where we reflect on how our findings fit into the current state-of-the-art.

Results

- For all figures that are relevant, please include how many cells were analysed in the legend.

We have now included the number of cells analyses in the respective figure captions.

- Although there might be space constraints, it would be helpful if wt panels could be inserted alongside perturbed panels (e.g. of flow topologies) to enable easier comparison.

We like the Reviewer's suggestion, but could not find a way of inserting wt panels in this way due to space constraints. Additionally, we feel that the resulting panel redundancy would be infelicitous, so we would like to suggest keeping the panel structure as it is.

- Throughout the manuscript, 'strain rates' are discussed in the text but the figure labels say 'strain'. These are two different measurements. Can this please be clarified.

We corrected the figure labels to "strain rates" where appropriate.

- Figures would benefit from more extensive labelling, especially for axes such as A-P and D-V. Marks to clearly indicate points of interest (e.g. flow collision points) would also be helpful

We have added more extensive labelling in figures throughout the manuscript as suggested by the Reviewer. We have reviewed all the image labels to be consistent.

- Fig 1K&I. I can't make out the details of the blue lines in I.

Blue lines indicate the histograms of individual cell orientations. We have now highlighted this in the figure legend.

- Supplementary figure 2a: axis labels would be helpful.

We have added an axis label to this figure.

- Line 167. I suggest that referencing the Araya paper (ref 45) detailing pMyosin and Myl12.1-GFP upregulation at the midline internalisation site in the hindbrain would be pertinent at this point.

We have now added this reference in the appropriate section.

- Supplementary video 2, figure 2a, supp fig 2b: Annotation would make it clearer. Also, in the video legend ventral should be down, not 'right'.

We have reannotated these figures and corrected the annotation in the video figure legend. We also made similar changes to figures and legends throughout the manuscript to remain consistent.

- Figure 2C. I am confused by these strain rate maps. Some of this may be remedied by fuller explanation in the figure legend and potentially combining supplementary figure 2d with figure 2c. It looks like the strain-rate heat map is overlying the velocity vector maps, similar to those shown in figure 1f. In that case, how have the AP and ML strain rates been separated on the PIV plot? I think the vectors should be corresponding to anisotropic strain rate and not velocity. If the tissue is internalising at this point, would this also give a similar strain rate map, without compression having occurred? I think the data and conclusions are fine – it just needs clearer presentation.

The time-averaged strain rates are indicated in the colour coded areas and the direction along the anterior-posterior (AP) and left-right (ML) axis have been calculated separately as outlined in the Material and Methods section. In contrast, the velocity vectors directly indicate the 2D tissue flow (as also shown in Fig. 1) to compare the local tissue deformations with the overall cell movements. To make this clearer, we have included more explanations in the corresponding figure legends.

In the cases where cells do not internalise such as for example in MZoepe or *cdh1* MO embryos, the compressions seen in wt embryos do not occur, indicating that deformations are linked to internalisation events.

- I am not able to assess the supplemental note describing the mathematical model, since this is outside my expertise, but the output shown in figure 3 is interesting and makes sense.

We thank the reviewer for appreciating our modelling approach.

- I would have found it easier to follow if the “Internalisation depends on E-cadherin-mediated tissue interaction” section was after the “Differential behaviours of mesendoderm cells initiate ANP tissue folding and cell internalisation” section.

We thank the reviewer for this suggestion. We agree that this might be a more logical order and have accordingly moved Fig. 6 up to become Fig. 3.

Discussion

- One thing that I did find a bit confusing throughout the manuscript was the overall message of how the flows and compression points linked to the folding and internalisation events. All the information is there in the manuscript, but it would benefit from a summary figure and/or conclusions section. Figures 1m and 2k are helpful. Perhaps some kind of combination of the two, showing the morphogenetic movements that occur in the ANP and how the flows link to this.

We have now revised the Results and Discussion sections to make it easier to follow the relationship between 3D reorganisation in the tissue including the link between flows and internalisation. We have also revised our schematics throughout the manuscript to illustrate that point better and have included an additional summary schematic in Supplementary Fig. 5k.

- I suggest that the manuscript needs a clearer explanation of what is meant by ANP folding. Is this really tissue folding (which is more usually associated with an already polarised epithelium) or is it more akin to ‘cell internalisation’, as described in reference 45? This could be incorporated into the suggested summary figure in the point above.

We thank the reviewer for mentioning this important point. Usually folding is associated with out-of-plane tissue deformations of polarised epithelial monolayers. We think that unlike in polarised epithelia, forces exerted on the ANP lead to more complex deformations within the bulk of the tissue which might induce buckling- and folding-like events. For this reason, we now refer to these deformations as “multilayer-folding” throughout the manuscript.

We have now investigated this further by tracing individual cells within one layer of the ANP and found that neuroectoderm cells usually remain in contact with their respective neighbours and reorganise in such a way as to define a folded shape oriented parallel to the initial plane (new Fig. 1n and Supplementary Fig. 1m).

We have given a more detailed response to this question including a simple model prediction in our response to Major Comment 3 of Reviewer 3 below, to which we refer the Reviewer. We have also discussed this in more detail in the revised Discussion section.

- Another thing that is currently missing is a fuller explanation of how ANP morphogenesis and the results of this manuscript fit within:

a) Zebrafish neurulation

- how is this early neural plate shape important for later neurulation? E.g. how does this early folding event relate to the later folding events described in the same tissue (e.g. reference 71 Brewster lab)?

- How does ANP morphogenesis relate to hindbrain neural plate morphogenesis (e.g. reference 45 Araya paper)? Why does it occur via a different mechanism (i.e. not via NMYII)?

b) Vertebrate neurulation

- How do the morphogenetic events described here relate to amniote neural plate morphogenesis and later neurulation?

- I suggest that the final sentence of the discussion section is a bit out of place and could be replaced by a stronger concluding sentence.

We thank the reviewer for pointing this out. We have now extensively revised our discussion section to address the questions raised above. We have also modified the end of the discussion and the final sentence. We think these additions have substantially improved the discussion section and highlight our findings better in comparison to other studies in the field.

Methods

- Can you indicate on figure legends which imaging modality was used?

We added the imaging modalities used to the figure legends where appropriate.

Reviewer #2 (Remarks to the Author):

In their manuscript, Inman et al analyze the morphogenesis of the anterior neural plate in zebrafish embryos using quantitative imaging and genetics. The anterior neural plate rests on top of the mesendoderm and folds during neurulation. The authors use a transgenic line to label the anterior neural plate and also label the membranes of the cells. This allows them to look at tissue movement/flow. They then developed an image software pipeline and projected the anterior neural plate in a 2d-plane and extracted the movement and shape changes. Using an approach from Richard Adams' group, they then calculate the anterior-posterior and medial-lateral strain rates. Together, these approaches show that the anterior neural cells move towards the midline and are compressed along the midline as the tissue folds. This leads the authors to propose a model in which the mesendoderm (that is located underneath the anterior neural plate) pulls on the neural cells in the front and restrains the neural cells in the back. The authors go on to model this process and find that their idea of how the mesendoderm pulls on the anterior neural plate can generate the observed flow patterns in silico. To support their model molecularly they assess whether the anterior neural plate adheres differentially along the AP-axis (they look at cell displacement as a measure for this) and whether F-actin levels differ along the AP-axis. They find that the front of the mesendoderm slides/moves along the neural plate while the back does not and that there is more Lifeact-GFP in the front and more gfp-tagged cdh1. They then perform functional assays. They block mesendoderm formation and find that the neural plate does not fold. They then place mesendoderm cells expressing dnRac1 in front of the anterior neural plate (these cells are assumed to be immotile) and find that mesendoderm cells do not move much and the neural plate does not fold. They also test the role of convergent-extension movements using wnt11 mutant and find that this does not affect folding. Lastly, they reduce the levels of cdh1 using morpholinos and find that adhesion between the neural plate and mesendoderm was reduced (judged again by cell-cell displacement) and folding was also affected. The strengths of this study are the nice quantitative image analyses (which are mostly well documented and convincing) and the weaknesses are the molecular experiments that do not provide much mechanistic insight into the forces underlying neural plate folding and are preliminary at the current stage. I therefore do not think that this manuscript merits publication in a high profile journal in its current form.

Major concerns.

1. The authors find that the mesendoderm interacts with the overlying neural plate differently in its front and back. One key insight would be to know the molecular nature of this difference – this is a tough question to answer but would definitely move this manuscript from a more descriptive to a more mechanistic study.

The focus of our story is the cell mechanical and tissue mechanical mechanisms that control early anterior neural plate (ANP) formation. We contend that these are very much mechanistic rather than descriptive results that show how local and global force production leads to the observed reorganisation of the ANP. In particular, at the molecular level, we identified the role of the adhesion receptor E-cadherin in regulating force transduction at the tissue interface to support cell local internalisation and global tissue reshaping. We do agree that exploring the molecular reason underlying the spatial differences in E-cadherin localisation that we discovered is an interesting question worth following up in future work, but we believe that this is beyond the scope of the current manuscript.

However, to further identify molecular regulators of the tissue remodelling process that we uncovered, we now provide additional new insights regarding the potential role of two key molecular players during gastrulation, Sonic hedgehog (Shh) and Nodal, in regulating internalisation. First, we inhibited Sonic hedgehog (Shh) signalling, which is produced in the axial mesendoderm and was shown to be important for DV neural plate patterning (Lumsden and Graham 1995). We found that formation of the ANP in shh inhibited embryos occurs similar to wt embryos and does not affect neuroectoderm internalisation (new Supplementary Figure 4p) or ANP reshaping (new Figure 5p and new Supplementary Figure 4q), suggesting a negligible role for shh in ANP morphogenesis during gastrulation. Next, we investigated a potential dual role for Nodal signalling independent of cell fate specification. Nodal has recently been shown to have a mesoderm-independent role in supporting neuroectoderm CE (Williams and Solnica-Krezel 2020) and also to modulate E-cadherin contact duration of mesendoderm cells (Barone V et al 2017). We found that while the ANP does not fully converge to the midline and remains slightly expanded laterally (new Figure 5q and new Supplementary Figure 4s), neuroectoderm internalisation occurs similarly to wt embryos when Nodal signalling is inhibited (new Supplementary Figures 4r), indicating that Nodal signalling is likely dispensable for neuroectoderm internalisation during gastrulation. Although other biochemical signalling cannot be ruled out entirely, these additional observations stress the importance of the mechanical mechanisms that we have discovered in controlling ANP reorganisation.

2. Image segmentation. From the images in the figures it is hard to judge how well the cell segmentation worked. They should quantify this. How many cells are correctly isolated? Also, it would be nice to know how the watershed segmentation was done. Did the authors use marker-controlled watershed segmentation?

We have included a section in the Material and Methods section in which we estimate the segmentation accuracy. Briefly, the mean absolute error in length measurements (135 cells measured from 3 embryos) as a proportion of the true length of the cell is 0.17 ± 0.19 (SD). The mean error in length measurements as a proportion of the true length of the cell is -0.02 ± 0.25 (SD). This suggests that the length measurements have a large range but are not biased in either direction. Similar, the mean absolute orientation angle error is 22.9 ± 21 (SD) and the mean angle error is 5 ± 31 (SD), suggesting no bias in

either direction. We found that there is no substantial change if separated by distance to the dorsal midline or time, with the possible exception of cells at 10 hfp.

The 2D watershed segmentation was performed using Matlab's watershed function, after preprocessing using Gaussian smoothing. No marker control step was used in the preprocessing.

3. Cdh1 analysis. Expressing *cdh1* from the *gsc* promoter does not report on levels of *cdh1* (the *gsc:gfp* is not a good control since it is a small transgene at a different insertion site). There are knock-in lines for *cdh1* available (PMID: 30969163). These lines should be used to assess levels and localization of *cdh1*. Also, overall levels are not key here. Key is how much *cdh1* localizes at the interface of the mesendoderm and the neural plate. In fact it is surprising that overexpression of *cdh1* in the mesendoderm does not affect the embryo.

We agree with the reviewer that the *gsc* promoter driven E-cadherin is not directly reporting endogenous E-cadherin levels. We also agree that the *gsc:GFP* insertion site is likely at a different location, but we believe that this would rather impact on the overall expression levels than the distribution between different cells. Our analysis focused more on such relative differences between populations of cells in the mesendoderm rather than these total levels of E-cadherin. Still, we agree with the Reviewer that localisation of E-cadherin between cells at the single-cell level would be more informative to estimate these disparities. We have therefore now quantified localisation of E-cadherin between neuroectoderm and mesendoderm front or rear cells. We found a similar trend, whereby E-cadherin is more enriched between rear mesendoderm and neuroectoderm cells (that eventually internalise) than between front mesendoderm and neuroectoderm cells (new Figure 2i), confirming our previous findings.

As our system does not directly report on endogenous E-cadherin levels, we also wanted to confirm our observations with endogenous E-cadherin as suggested by the Reviewer. Despite continuous efforts, we were not able to use the *cdh1* knock-in line to answer this question due to problems related to importing fish from overseas in the time frame of the revision. Nevertheless, we reasoned that we could also visualise endogenous distribution of E-cadherin using immunofluorescence staining. In these experiments, we found stronger E-cadherin accumulation between rear mesendoderm and neuroectoderm cells than between front mesendoderm and neuroectoderm cells (new Figure 2j, k and new Supplementary Fig. 2i). These findings also establish that the *Tg(gsc:cdh1-EGFP)* line that we generated is a valuable tool to visualise internal E-cadherin dynamics. These results, together with the differences in measured contact duration (Fig 2e, f), indicate local differences in adhesion between the two tissues which likely originate from a more mesenchymal-like front and an epithelial-like rear of the mesendoderm tissue. We add that similar observations have been made in other migrating collectives such as between leading and trailing cells in the lateral line primordium in zebrafish.

4. Lifeact analysis. Similar to *cdh1*, total levels of lifeact-gfp are meaningless. Those levels depend on the promoter (b-actin in this case) and not f-actin levels. The authors need to look at subcellular localization and ask whether there is more localized lifeact in the cells in the front and rear.

To address this, we first quantified the subcellular localisation of lifeact-GFP in mesendoderm front and rear cells located at the tissue interface. We found that cells at the front of the collective had enriched lifeact-GFP levels at the leading edge compared

to cells at the rear (new Figure 2n), indicating that front cells are more likely to form F-actin rich protrusions. To confirm this, we counted the number of protrusions in front and rear cells produced over time. We detected that significantly more protrusions (3-fold difference) are formed in front than rear cells (new Figure 2m), suggesting that front cells are highly migratory.

5. Transplantation. The authors need to do control transplantations. Wt into wt – does placing cells in front of the neural plate affect morphogenesis.? Also, the clones should be positively marked, ie RFP expression and not absence of RFP expression in an RFP-expressing host. Absence is impossible to see/score. Also, the authors need to confirm that dnRac1 expression affects actin dynamics and cell movements as they assume.

We have performed the requested control experiments and checked whether transplanted mesendoderm wt control cells alone would affect ANP morphogenesis. It has recently been shown that transplanted mesendoderm wt cells move directionally once they join the endogenous mesendoderm collective (Dumortier JG et al 2012) In agreement with these observations, we found that transplantation of wt control cells does not interfere with endogenous mesendoderm migration compared to transplanted dnRac expressing cells (new Supplementary Figure 4k). Transplanted wt control cells also displayed much higher levels of protrusive activity than dnRac expressing cells indicating that F-actin dynamics are perturbed in dnRac expressing cells (new Supplementary Figure 4j). Further, neuroectoderm cells internalised at a comparable rate in wt control transplanted embryos, in contrast to dnRac expressing transplanted cells (new Supplementary Figure 4l).

In our experiments, host and donor cells are clearly distinguishable by using two fluorescent markers for the host (RFP, GFP) and one fluorescent marker (GFP) for the transplanted cells. This enabled us to distinguish these two cell populations.

6. The model the authors put forward argues that the mesendodermal cells in the front migrate forward pulling on the overlying neural cells and pushing them backward. The rear mesendodermal cells – in contrast – are held back by the overlying rear neural cells. So, the mesendodermal cells in the front are pulled forward while the mesendodermal cells in the back are held back. Since the tissue does not snap in two, mesendodermal cells need to fill in or mesendodermal cells need to be stretched (increase in strain rate). Do the authors see such kind of cell behavior in the mesendoderm?

From the model that we propose, it does not follow that the mesendoderm needs to stretch. It suffices for the overlying neuroectoderm cells to be compressed as a result of the differential forces exerted by the mesendoderm, which is indeed what we observe.

We add that all cells in the mesendoderm collective are moving anterior during gastrulation. Yet, there are stage-dependent differences in the velocities within the collective along the anteroposterior direction of the tissue. To demonstrate this, we quantified the instantaneous velocities of mesendoderm front and rear cells between 7.5-9 hpf (new Supplementary Figure 2g). We found that the velocities of front cells decrease over time which is consistent with the notion that mesendoderm migration halts at the end of gastrulation. In contrast, rear cells speed up over time and eventually exceed front cell velocities, in agreement with a contribution of posterior CE such as in the notochord, thereby pushing the prechordal plate mesendoderm anterior. These differences in local movement behaviours over time would support the integrity of the mesendoderm and prevent tissue disintegration.

Minor

1. The authors use neural cell to mesendoderm cell neighbor distance as a measure of adhesion. This could be correct but it could also reflect different degrees of mobility so strictly speaking this is not a direct measure of adhesion.

We measured the duration of the time that a migrating mesendoderm cell stays in contact with the overlying (non-migrating) neuroectoderm cell as a function of the position of the cells within the mesendoderm collective. We agree with the Reviewer that different degrees of mobility could in principle achieve similar differences in time durations if front cells consistently migrate faster than rear cells. However, as addressed in point 6 above, the opposite is true in our system: mesendoderm front cells slow down over time whereas rear cells speed up relative to each other (new Supplementary Fig. 2g). This suggests that the binding duration is not linked to mobility differences and we conclude that the binding duration represents weaker or stronger binding affinities of the mesendoderm cells to the neighbouring tissues.

Reviewer #3 (Remarks to the Author):

The manuscript by Inman et al. describes the cell and tissue shape changes that occur in the anterior neural plate of Zebrafish during gastrulation. The authors first examine changes in the shape of the anterior neural plate using the Tg(*otx2::Venus*) reporter. They then map tissue flows across the embryo using PIV, which identified planar rotational flows and convergence and extension movements. Modeling of neuroectoderm tissue flows predicted that mechanical transitions at distinct stages can lead to observed flows. The authors then performed experiments removing or preventing migration of mesendoderm to suggest a role for mesendoderm migration in neuroectoderm internalization. Finally, they showed that internalization dependend on E-Cadherin, but not convergence and extension or N-cadherin. Overall, the manuscript described the early shape change of the anterior neural plate, but I did not find the mechanistic conclusions about tissue shaping to be well supported by the data, as I describe in my point-by-point response.

Main Comments:

1) Lines 86-108: The authors conclude that the anterior neural plate is dramatically reshaped throughout gastrulation. An alternative model is that there are changes in gene expression for the *otx2::Venus* reporter. The tracking of flows with PIV does not address this point, because individual cells and their *otx2* expression status are not tracked and the third dimension is not accounted for. The authors should comment on previous data or show how the status of *otx2::Venus* expression changes (or doesn't) during this reshaping process.

To address this, we tracked individual cells (nuclei) in the anterior neural plate during gastrulation and compared the movements with the *otx2::Venus* expression levels over time (new Supplementary Fig. 1c). We found that the *otx2::Venus* signal increases over time in cells that become neuroectoderm-fated and remains on during gastrulation. In contrast, presumably non-neuroectoderm cells (with very low background levels of *otx2::Venus*) also move, but do not increase in *otx2::Venus* levels. These observations

therefore indicate that the alternative model whereby the appearance of the neural plate is determined by gene expression and fluctuating *otx2:Venus* levels is unlikely.

2) Lines 139-140: The authors say that the convergence of lateral cells results in cell internalization that ‘resembles tissue folding’. However, the images in Fig. 1i look more like a thickening of the tissue rather than a fold (there is no indentation at the surface). This stage does not seem to approximate the tissue folding that occurs later in neurulation, which was shown to have similarities with primary neurulation in mammals (Werner et al. 2021). I disagree with the use of ‘tissue folding’ throughout this paper, that language is misleading and creates a false analogy to other tissue folding processes.

We thank the reviewer for raising this important point. Folding is usually associated with polarised epithelial monolayers where in plane forces lead to out of plane bending of a sheet and the way we use the word folding in our context might be misleading. The ANP is a multilayered tissue deforming under the action of multiple forces that might result in more complex buckling and folding events than those observed in a monolayer. To clarify the tissue deformations in the bulk of the internalising tissue, we manually tracked cells within the ANP along the interface to the underlying mesendoderm (new Fig. 1n; Supplementary Fig. 1m). We found that neighbouring cells usually keep their respective neighbours and reorganise in such a way as to define a folded shape. Notably, multiple stacked folds can be formed from such a layer during internalisation. These are oriented somewhat parallel to the plane of the initial layer (asymmetrical folding), rather than perpendicular (symmetrical) to it, as usually seen in folded epithelial monolayers. Moreover, as cells move continuously anterior during this process, this folding need not give rise to any notable indentation of the surface. Interestingly, when the bulk of the tissue movement is displaced ventrally without anterior extension, as in the case of the *wnt11* mutant, we do observe surface indentation of the embryo, which might indicate a more symmetrical folding (Supplementary Fig. 6j). To emphasise that the deformations in the ANP bulk differ from classical epithelial folding, we refer to them now as “*multilayer folding*” throughout the revised manuscript.

We are now working on a mechanical description of this multilayer folding that is not captured by our current theoretical model (the purpose of which is to describe the tissue flows in the plane of the ANP surface). Since the cell tracking suggests that there are few, if any cell intercalations in the folded surface, an elastic description of this multilayer folding is appropriate. Here, we therefore offer a minimal elastic model of multilayer folding: We consider an inextensible elastic rod, initially horizontal, representing (part of) the initial, uninternalised surface of the ANP. The end-to-end distance of the rod is reduced (representing ingression of cells) and a horizontal force is applied to the rod (representing pushing due to mesendoderm migration and convergent extension). This buckling problem is different from the classical rod buckling problem, in that the force is applied not only to the ends of the rod. This breaks the symmetry of the buckled shape, and indeed leads to folded shapes (Figure R1 below) reminiscent of those seen in the cell tracking data of the revised manuscript. Below we added a simple model description of this problem.

Minimal model of multilayer folding

The ends of an inextensible elastic rod of unit length are clamped into a horizontal plane. The end-to-end distance of the rod is reduced to $1 - d$, for some displacement $d > 0$, and a horizontal force F is applied to each point in the rod. The resulting, deformed shape of

the rod is $(x(s), y(s))$, and $\psi(s)$ denotes the tangent angle to the rod, where $0 \leq s \leq 1$ is arclength. The Lagrangian governing the deformations is

$$\mathcal{L} = \int_0^1 \frac{\dot{\psi}^2}{2} ds - \nu \int_0^1 \sin \psi ds - \int_0^1 \lambda(s)(\dot{x} - \cos \psi) ds,$$

in which dots denote differentiation with respect to s and where the first term is the bending energy of the rod, the Lagrange multiplier ν in the second term imposes the condition that the ends of the rod lie in the same horizontal plane, and the Lagrange multiplier function $\lambda(s)$ imposes the geometric relation $\dot{x}(s) = \cos \psi(s)$. The variation of this Lagrangian is

$$\delta \mathcal{L} = \left[\dot{\psi} \delta \psi - \lambda \delta x \right]_0^1 + \int_0^1 [(-\ddot{\psi} - \nu \cos \psi - \lambda \sin \psi) \delta \psi + \dot{\lambda} \delta x + (\dot{x} - \cos \psi) \delta \lambda] ds.$$

Meanwhile, the infinitesimal work done by the horizontal force is

$$\delta \mathcal{W} = \int_0^1 F |\sin \psi| \delta x ds,$$

wherein the factor $|\sin \psi|$ ensures that only component of the horizontal force normal to the rod is felt. Imposing $\delta \mathcal{L} = \delta \mathcal{W}$ yields the governing equations

$$\ddot{\psi} = -\nu \cos \psi - \lambda \sin \psi, \quad \dot{\lambda} = F |\sin \psi|, \quad \dot{x} = \cos \psi,$$

to be solved subject to the boundary conditions

$$\psi(0) = \psi(1) = 0, \quad x(0) = 0, \quad x(1) = 1 - d, \quad \int_0^1 \sin \psi ds = 0,$$

imposing respectively the conditions of clamped ends, the end-to-end displacement condition, and the condition that the ends of the rod lie in the same horizontal plane. These equations are easily solved numerically using the boundary problem solver `bvp4c` of MATLAB (The MathWorks, Inc.).

Figure R1. Minimal model of multilayer tissue folding. The end-to-end distance of an inextensible elastic rod (solid line, representing part of the dashed initial surface of the ANP) is reduced (representing ingression of cells) and a horizontal force is applied to it (representing pushing due to mesendoderm migration and convergent extension), in the direction indicated by the arrow. The resulting buckled shape is asymmetrically folded, in a manner reminiscent of the multilayer folding observed in the ANP. A \leftrightarrow P: anterior and posterior.

3) Fig 2 and Supplemental Figure 2: When the authors see cell stretching, how do they know they are seeing the whole cell in each time step? Would be stronger evidence to show 3D reconstruction of entire cell.

We have now included 3D reconstructions as suggested by the Reviewer; these are described in detail in the Material and Methods section. We have also included representative images of these 3D reconstructions in new Fig. 2a and Supplementary Fig. 2c. We further repeated our cell deformation measurements based on these new 3D shapes. Briefly, the resulting measurements including cell lengths and cross-sectional areas calculated from the 3D cell reconstructions show the same trend as in the 2D measurements (new Supplementary Fig. 2d), but with reduced accuracy in later frames (as noted in the Material and Methods section). This may be in part due to the cells from those frames being longer and thinner and therefore harder to segment. Overall, we

conclude that the 2D measurements are a good proxy in our system to measure cell length.

Also, cell stretching seems to happen after the cell has internalized, which is inconsistent with the language in Lines 169-172.

This might be a misunderstanding. We refer here to deformation of cells once internalised. We have rephrased this in the text to make this clearer.

4) Lines 342-344 and Fig.4: An alternative model to the mesendoderm providing a force-generating role is that the mesendoderm provides a biochemical signal and that the position of this signal is important for proper internalization. The brute force approach of removing or preventing mesendoderm movement do not rule out the signaling model or definitively shown the importance of force generation.

The focus of our study is the cell mechanical and tissue mechanical mechanisms that control ANP morphogenesis, which emphasised particularly, on the molecular side, the function of the cell-cell adhesion molecule E-cadherin in heterotypic tissue coupling. This study is agnostic to the potential molecular signals that set up this observed differential adhesion and the internalisation (and hence set off the tissue deformations of which we elucidate the mechanical mechanism). In particular, if internalisation were caused by a biochemical signal that needs to be positioned correctly (as hypothesised by the Reviewer), our *in vivo* and *in silico* perturbations still demonstrate the importance of the correct mesendoderm deformation and hence of the underlying mechanical forces and, by extension, of the force-generating mechanism that underpins them.

Still, we agree with the Reviewer that the exploration of these signals, beyond the scope of this study is a worthwhile direction for future work. To address this, we have now investigated the role of two morphogens, Sonic hedgehog and Nodal, that have previously been shown to impact on neural plate development (as also highlighted in our response to Reviewer 2). First, we inhibited Sonic hedgehog (Shh) signalling, which is produced in the axial mesendoderm (notochord) and was shown to be important for DV neural plate patterning (Lumsden and Graham 1995). We found that formation of the ANP in Shh inhibited embryos occurs similar to wt embryos and that neuroectoderm internalisation (new Supplementary Figure 4p) and ANP reshaping (new Figure 4p and new Supplementary Figure 4q) are not affected, suggesting a negligible role for Shh in ANP morphogenesis during gastrulation. Next, we investigated a potential dual role for Nodal signalling independent of cell fate specification. Nodal was recently shown to have a mesoderm-independent role in supporting neuroectoderm CE (Williams and Solnica-Krezel 2020) and also in modulating E-cadherin contact duration of mesendoderm cells (Barone V et al 2017). We found that in embryos in which Nodal signalling is inhibited, the ANP does not completely converge to the midline and remains slightly extended laterally (new Figure 4q and new Supplementary Fig. 4s). However, neuroectoderm internalisation occurs similar to wt embryos when Nodal is inhibited (new Supplementary Figures 4r), indicating that Nodal signalling is likely dispensable for neuroectoderm internalisation during gastrulation. Although other biochemical signalling mechanisms cannot of course be ruled out entirely, these additional observations stress the importance of the mechanical mechanisms that we have discovered in controlling ANP reorganisation.

5) Supplemental Fig. 5 and Fig. 6: The authors say that heterotypic interactions between N- and E-cadherin underlie the interaction between neural plate and mesendoderm.

Thus, if a force-based mechanism existed for mesendoderm to internalize neuroectoderm, it would require both N- and E-cadherin. The fact that *cdh2* morphants did not affect internalization argues against the authors point.

We believe that the reviewer's comment is likely based on a misunderstanding. In the manuscript we refer to heterotypic tissue interactions such as between mesendoderm and neuroectoderm, and not heterotypic cadherin interactions (e.g. between E and N-cad). We thank the reviewer for pointing this out; we have clarified this point in the main text to avoid confusion.

The *cdh1* morphants likely affects many tissue types and doesn't necessarily argue for a direct connection between mesendoderm and neuroectoderm.

We have previously shown that moderately reducing E-cadherin levels in the embryo affects force coupling between mesendoderm and neuroectoderm and perturbs their coordinated movements (Smutny et al., 2017) without significantly affecting gastrulation. The reviewer is correct that this does not allow us to entirely rule out effects on other tissues such as interactions between the neuroectoderm and the EVL as described in *cdh1* mutants (Babb and Marrs, 2004). However, we still expect that mesendoderm-neuroectoderm tissue interaction is the most relevant one for the mechanical coupling required in the mechanism that we propose in this study to drive correct tissue flows and internalisation. To test whether local tissue-specific perturbation of E-cadherin-mediated adhesion is sufficient to affect internalisation, we therefore transplanted *cdh1* morphant mesendoderm cells into the rear of the endogenous mesendoderm collective where internalisation occurs (new Figure 3l). We found that local weakening between rear mesendoderm and overlying neuroectoderm was sufficient to perturb internalisation and neuroectoderm cells remained more superficial (new Figure 3m, n), similarly to our observations in *cdh1* morphant embryos. These results strengthen the role for E-cadherin in specifically regulating the inter-tissue coupling necessary for tissue flows and internalisation.

Minor Comments:

1) Supplemental Fig. 2b, c: The cells in the region look multilayered and there are no polarity markers being used, so what enables the authors to define apical vs. basal?

To avoid confusion with polarised cells, we now use the words "top" (closer to the embryo surface) and "bottom" (further away from the surface) area of the cell.

2) Line 164-167: If internalization is stepwise as shown in Fig. 1j, then why would the authors expect that myosin II would be regionally enriched? It could be enriched in single cells that undergo internalization.

Our findings show that neuroectoderm internalisation occurs stepwise in the AP direction over time, but that cells internalise collectively along the ML direction close to the dorsal midline (Figure 1i, j). To test whether internalised cells along the ML axis express increased levels of myosin 2, we quantified phospho-myosin 2 levels at the single cell level. We found that phospho-myosin is predominantly localised at cell junctions and that there is no clear pattern of cells with upregulated myosin 2 activity evident across different embryos (new Supplementary Figure 2b). In particular, we did *not* see a pattern of myosin

being enriched in potentially internalising cells at the midline. Instead, cells with higher myosin 2 activity seem to be distributed in a random fashion in the ANP, which cannot easily explain the internalisation behaviour observed. Hence, we believe that an intrinsic force generating mechanism is less likely to drive the observed spatiotemporal coordinated internalisation, although we cannot completely rule this out.

3) Line 186: The authors argue that tissue compaction acts as a hinge point, but there is no bending at the surface of the tissue in the movies or figures. This language is imprecise and misleading, I suggest the authors reword it.

It looks like tissue compaction is leading to tissue thickening, not hinge point formation.

Tissue compaction occurs locally in the superficial layers and along the ML axis at the anterior most part of the ANP. Thickening occurs throughout the tissue all along the dorsal midline once cells internalise, but not in the area of compaction, which occurs ahead of the initial internalisation site. Internalised cells revolve around this compacted area which we described as “hinge point” (Fig. 2a, c). We understand the reviewer’s concern that this analogy could be misleading and associated with later appearing hinge points during neurulation, which are mechanistically different and occur in a single-cell-layer context. We reworded this in the text to clarify this point. Regarding the bending of the surface, we refer to our response to the Reviewer’s Major Comment 2 above.

4) Fig. 2j: The actin in the image doesn’t look cortical. Has it been processed (smoothed) in some way that was not described in the figure legend?

The lifeact-EGFP illustrates the average distribution of actin within the mesendoderm from a multiple slice projection colour coded for intensities, which might produce some smoothing. We have replaced this figure with a single slice to better highlight actin distribution in single cells at the ANP interface (Figure 2l).

5) The term multi-tiered is never explained in the manuscript

We have now explained the term “multi-tiered” in the main text, which simply refers to the multiple mechanical “steps” regulating ANP remodelling.

REVIEWER COMMENTS

Reviewer #1 (Remarks to the Author):

The authors have addressed my comments. Regarding figure 1K&L, I meant that I couldn't see the blue line histograms due to the small size of the figure. This is a minor point, but the clarity of the figure could be improved here.

We thank the reviewer for the very positive response regarding our revised manuscript. We have now separated the histogram (blue) from the average (red) and increased the size to improve visibility.

Reviewer #2 (Remarks to the Author):

In their revised manuscript, Inman et al. addressed a few of my concerns but the main issues (lack of mechanistic insight and lack of convincing/well-controlled experiments) were not addressed. As said initially, the quantification part looks solid but the molecular aspect of the study is preliminary/not convincing and potentially flawed. I therefore do not support publication of this manuscript. See details below and comments for the initial submission.

We appreciate that the reviewer acknowledges our efforts of the revised manuscript in part. We thank the reviewer for their continuous feedback, however, we disagree with some of the remaining comments as outlined in our point-to-point response below. We believe that the remaining concerns do not change the main observations and conclusions of this study and raise views that we believe are beyond the scope of this manuscript. As outlined in our previous response, the focus of this story are the cell and tissue mechanical mechanisms that control early anterior neural plate (ANP) formation. We argue that these are very much mechanistic rather than descriptive results that show through experimental and theoretical work how local and global force production leads to the observed reorganisation of the ANP. At the molecular level, we explored the role for E-cadherin and N-cadherin receptors, and the key morphogens Sonic hedgehog and Nodal in regulating ANP tissue reshaping. Our results indicate a minor role for these morphogens in controlling neuroectoderm internalisation, but instead highlight an important role for the adhesion receptor E-cadherin in regulating force transduction at the tissue interface to support local cell internalisation and global tissue reshaping. Investigating the molecular process underlying the spatial differences in E-cadherin localisation in mesendoderm cells that we discovered is an interesting and complex question worth pursuing in future work (as debated in the discussion section), but we believe, is beyond the scope of the current manuscript.

Regarding the specific remaining comments from the reviewer, please find our responses below.

1. Cdh1 analysis.

There are a few issues with the authors analysis/argumentation.

- Gsc:cdh1-gfp. The expression levels of cdh1-gfp depend on the promoter gsc and the insertion site of the transgene, so any differences in expression are reporting on the promoter activity in the different cells and the effect of the insertion site and not on any differences in cdh1 expression. As I said this is pretty much meaningless.

- Effect of insertion site on transgene expression. Insertion sites do not only affect levels of transgene expression but expression also varies between cells depending on the insertion site. This is an unfortunate fact that makes the quantification of the expression of small transgenes pretty useless.

We agree with the reviewer that the insertion site can change expression levels of the transgene. Normally, we and others have not observed variegation of gsc expression levels between the cells within the same population due to the locus effect of the insertion in different zebrafish gsc transgenic lines that have been used extensively in the past (e.g. doi.org/10.1002/dvg.20253; doi: 10.1126/sciadv.abc5546; doi.org/10.1016/j.devcel.2022.05.001; doi: 10.1016/j.celrep.2016.06.036; doi: 10.1038/ncb3492). In particular, the specific distribution pattern of cdh1 in the mesendoderm that we consistently observe in embryos of the gsc:cdh1 transgenic line across different generations, we believe, is unlikely to be caused by insertion site effects across the cell population. This view is supported by similar localisation patterns that we observe with endogenous cdh1, where expression levels could potentially differ as the reviewer points out, suggesting that expression levels alone are likely not sufficient to explain these differences. In this study we focused on the localisation/accumulation of cdh1 between mesendoderm and neuroectoderm cell-cell contacts which does not necessarily reflect total protein expression levels. In fact, multiple mechanisms regulate cdh1 surface levels and recruitment such as intracellular trafficking, transcriptional regulation, post-translational modification, junctional actomyosin contractility and biochemical signalling pathways downstream of cadherin including associated proteins (doi: 10.1038/ncb3136, doi.org/10.3389/fcell.2021.701175). The molecular underpinning of cdh1 recruitment to heterotypic cell junctions in our system is an interesting and complex question, but we believe beyond the scope of the current manuscript.

- Cdh1 antibody staining. This is a much better approach but the authors should show a complete image of the front and rear from the same embryo. Antibody stainings vary a lot between embryos. About 5-fold within a batch of embryos stained together. So an image similar to Fig. 2g is needed. The Fig. S2i is split and does not help.

The image in question showed the front and rear in the same embryo. To achieve optimal magnification for cellular resolution of cdh1 at the interface, only the front and rear were imaged and not the region in between. An overview of the full tissue was captured at a lower magnification, which has now been included for clarity (Fig. 2g; Suppl. Fig. 3a), together with high magnification images (Fig. 2g) and quantification of the interface at single cell contacts (Fig. 2h).

- Cdh1 knock-in lines. The timeline for revision set by the editors should allow for the import of lines. This is more of a comment for the editors than the authors. Since the manuscript was initially submitted 4 months ago, there was ample of time to receive the lines. So I do not think this is a valid excuse for not doing the experiment.

As this was addressed to the editors, we leave this for the editors to comment on.

- The differences in Cdh1 levels seen in Fig. 2g are likely due to the gsc promoter activity at the given genomic insertion site and not due to differences in cdh1 promoter activity (since it is the gsc promoter, the knock-in would address this) or altered cdh1 localization within the cell (levels are fairly the same along the cell membranes in the antibody stainings).

Differential turnover, as the authors point out, is possible but to make that claim they need to use the endogenous promoter to exclude differences in promoter activity.

We see a clear difference in the localisation of immunostained cdh1 at neuroectoderm-mesendoderm cell-cell contacts dependent on their position in the collective along the tissue

interface as shown in our previous images and in the newly provided Fig. 2g. In fact, our quantifications of endogenous cdh1 localisation at single mesendoderm-neuroectoderm cell-cell contacts (Fig. 2h) confirm this observation that cdh1 is enriched between rear mesendoderm cells and neuroectoderm cells.

- Minor. In my opinion the statement that “These findings also establish that the Tg(gsc:cdh1-EGFP) line that we generated is a valuable tool to visualise internal E-cadherin dynamics” is incorrect. I would not see why anybody would use this line over the knock-in lines. This line is creating a completely artificial scenario with cells mis-expressing cdh1. Mis-expressed cdh1 will not reflect the endogenous cdh1 and will likely also not resemble the endogenous protein dynamics and may cause unwanted phenotypes.

Quantification of cdh1 antibody staining indicate that the localisation pattern at mesendoderm-neuroectoderm cell-cell contacts along the tissue interface is similar under tissue-specific and endogenous expression. Differences in cdh1 expression levels would not explain such accordance. As outlined above, we believe that these differences are rather regulated on the protein level such as by recruitment or stabilisation of cdh1 at cell-cell contacts. The knock-in line will be valuable tool in the future to confirm these observations. However, studying cdh1 recruitment specifically in the mesendoderm, as we did here, is impossible to visualise with a cdh1 knock-in line, as cdh1 is labelled in both the epiblast and hypoblast.

2. LifeAct analysis.

- The quantification in Fig. 2n shows that both the front and rear cells have enriched LifeAct-GFP in their fronts. So, this is not supporting the authors hypothesis...

- More importantly, I have seen many LifeAct-FP images and movies and one normally sees nicely the outline of the cell cortex and cellular protrusions. I cannot see this in Fig. 2l. I would have guessed this is an image of a cytosolic FP. So much better images are needed.

- The authors claim that there are protrusions and provide a quantification in Fig. 2m but the primary movies of protrusions are missing. Since none of the images shows protrusions this is needed.

We thank the reviewer for pointing out the lack of clarity and provide now better images and graphs that should clarify these ambiguities.

We have now provided additional images of Lifeact-GFP expressing mesendoderm cells at the tissue interface including overviews of F-actin localisation in mesendoderm front and rear cells (Suppl. Fig. 3g), and high magnification images of representative cells in Fig. 2i. We have further included a graph where we quantified the number of total protrusions over time that we could identify (Fig. 2j) and intensity profiles of the cell outlines indicating F-actin enrichment in protrusive front cells versus F-actin localisation along the cell periphery in rear cells (Fig. 2k). Together, this demonstrates clear differences in protrusive activity depending on their location within the tissue along the neuroectoderm interface suggesting that front cells behave like highly polarised and migratory cells compared to rear cells, which typically do not show these features.

- As stated before, the intensity profile of LifeAct-GFP reflects reporter activity differences between the front and rear, not F-actin polymerization differences. This requires subcellular quantifications in Figs. 2l, S2j, S2k.

We have used lifeact-GFP to identify F-actin localisation and enrichment of F-actin in polarised protrusions of migrating cells, or absence thereof, as demonstrated in many previous studies investigating actin dynamics (doi.org/10.1038%2Fnmeth.1220);

[doi/10.1126/science.1224143](https://doi.org/10.1126/science.1224143); <https://www.nature.com/articles/s41467-024-47236-1>). We have not intended nor stated to measure differences in F-actin polymerisation. We measured protrusive activity of cells and F-actin localisation along cell periphery in single mesendoderm cells (Fig. 2j, k), which show clear differences in polarised protrusion formation of front vs rear cells.

3. Transplantation experiment (Fig. 3l–n).

- The authors decided not to address my concern that donor and host cells need to be labeled in different colors. In the *cdh1* mosaic analysis, the donor cells are labeled with membrane-bound GFP (GFP-CaaX). The highlighted cells do not show a membrane labeling although they should. The host cells are labeled with H2A-mCherry. However, nuclear labeling in the host cells is not discernable and the red channel is also shown as a gray scale (as is the green channel so one does not know what is what). The host cells are also labeled with cytoplasmic GFP so all host cells should be evenly green (or white given that the authors chose a gray LUT for both channels). This is not evident in the images. So, based on these data, the conclusions are not supported. See my initial comments.

For Fig 3n we chose a gray LUT, not to interfere with the colour-coded cell tracks. To outline the different labels and cells, we have now provided additional images in Suppl Fig 4n, showing all the single colour channels produced and a merged version. Due to imaging some cell populations in the same colour with different fluorescent intensities, some cells might show an oversaturated signal (e.g. membrane bound *gsc*-GFP-CAAX signals bleeding into the cell appearing cytoplasmic).

- Similarly, in the *dnRac1* mosaic analysis, the donor and host cells are labeled with a memGFP and the host cells express RFP (that oddly seems to be membrane bound). As said before, the clones need to be labeled with a FP. Absence of expression is not suitable for many obvious reasons – the main one is that no tg or injected mRNA is expressed in all cells so there will always be unlabeled host cells.

Host cells express membrane bound RFP (mRFP). To improve clarity, we decided to replace Fig. 5i with adequate images and also provided single colour and merged images in Suppl. Fig. 5h where the cells are easier to distinguish. Although we sort embryos and cells before transplantations to ensure cell labelling, there is no guarantee that 100% of all cells are labelled with the injected mRNA. However, the potential occurrence of occasionally unlabelled cells is not relevant for our analysis, as we only quantify (a fraction of) cells that we can clearly identify.

- Quantification of the absence/reduction of protrusions in *dnRac1* cells in Fig. S4j. The data need to be shown. Since none of the images/movies show protrusions, this is essential. We have now provided images of transplanted *dnRac1* expressing cells and wt transplanted cells in Suppl Fig. 5j.

Reviewer #3 (Remarks to the Author):

This manuscript details a complex morphogenetic process in vertebrates, showing how interactions between multiple tissues sculpts the anterior neural plate. The authors did a nice job addressing my comments and I just have some minor comments regarding clarity of the manuscript.

We thank the reviewer for the very positive response regarding our revised manuscript. Below we clarify the remaining minor comments below.

Line 22: The use of the word 'novel' could be misinterpreted. This is novel for the zebrafish ANP, but adhesive mechanisms resulting in friction is a general mechanism of morphogenesis.

We agree that friction through cell-cell adhesion has previously been observed during tissue morphogenesis. In this sentence we refer specifically to the here newly discovered mechanical mechanisms of shaping the zebrafish ANP, where friction is one of the key players amongst others. The adhesive mechanisms between heterotypic tissues generated by friction in the context of ANP morphogenesis has been explored elsewhere (Smunty M et al. NCB 2017). We hope that this clarifies the statement.

Line 218: typo: enablingt  enabling

Line 427(442): typo: contorl  control

We thank the reviewer for spotting this, we have corrected these typos in the manuscript.

Line 590, ref 58: I believe the authors have the wrong reference. The paper they cite is a Drosophila paper. I think the meant PMID: 34707290. If they are looking for Drosophila papers to make this point they could consider PMID: 28504247 and 36724258

We thank the reviewer for spotting this typo. The reference is indeed PMID: 28504247 (Ref 82).

Figures: In general it would be helpful to have embryonic axes specified throughout the figures, similar for what was done in Supp Fig. 5K

We thank the reviewer for pointing this out and have implemented more rigorous labelling as requested. Additional axes have been added to the following panels: Figure 2c, j, o; Figure 3h, j, l; Figure 5f, j, k, n; Figure 6f, h; Supplementary Figure 1h, m; Supplementary Figure 2f; Supplementary Figure 3k; Supplementary Figure 4d, n; Supplementary Figure 5h, j.

Reviewer's comments in **black**, author's response in **blue**.

Reviewer #2

1.) Cdh1.

Cdh1 knock-in line. This comment was addressed to both the editors and the authors. The authors chose not to get the line, which is odd because this would be the perfect experiment. The Cdh1 AB staining is ok but the localization to the membranes in Fig. 2g or Fig. S3a is not visible – one cannot see the membranes. So, this claim is not supported.

We have previously provided Fig 2g and Suppl Fig 3a as low magnification images (20x objective) to show endogenous cdh1 localisation along the whole interface as one continuous image as requested by the reviewer. We agree that cell outlines are harder to see in those low magnification images than in the high magnification images (40x objective) that we have already provided earlier. However, high expression of gsc:GFP-CAAX levels in mesendoderm cells clearly indicates the tissue boundary (Suppl Fig 3a). *We have now added an additional phalloidin staining of actin (previously not included) that also demarks the mesendoderm-neuroectoderm tissue boundary (Suppl Fig 3a).*

Moreover, we have previously provided high magnification images (40x objective) where we zoomed in on those regions (hence separate images were created but taken from the *same* embryo) that clearly showed endogenous cdh1 localisation at cell-cell contacts (previous revised version Fig 2j and Suppl Fig 2i), which we based our quantification on (current Fig 2h). We cannot reach this level of detail with the 20x images. We would therefore suggest incorporating both 20x as overview and 40x for more detailed images in the manuscript to avoid ambiguities. *We have now moved the 20x low magnification images including an overview of cdh1 localisation along the mesendoderm-neuroectoderm interface to Suppl. Fig. 3a and b. We have reintroduced higher magnification images (40x) of the front and rear (taken from the same embryo) in Fig. 2g and additionally show single channels in Suppl Fig 3c.* This includes endogenous cdh1 localisation at mesendoderm-neuroectoderm cell-cell contacts in addition to gsc:GFP-CAAX signal in mesendoderm cells demarking the tissue boundary.

We currently do not have the cdh1 knock-in line in place to perform additional checks on cdh1 localisation. However, although localisation of endogenous cdh1 (and expressed in the mesendoderm only) supports that adhesion between mesendoderm and the ANP is directionally dependent, localisation studies per se do not establish a fully functional role. Importantly, we have provided evidence for a functional role of cdh1 in perturbation assays in Figure 3 and Suppl Fig 4, where we interfered (globally and locally) with cdh1 function in the embryo. For us, these experiments provide the strongest support for a functional role of cdh1 in ANP reshaping by supporting inter-tissue cohesion and force transduction.

However, Fig. S3a clearly shows more Cdh1 in the ectoderm than the mesoderm. Whether this difference in Cdh1 levels demarcates the neuroectoderm-mesoderm border/tissue interface is impossible to say from Fig. S3a (the dotted line the authors draw is partly going through nuclei, which is likely not the border...).

Immunostainings rely on antibody penetration of the embryo. Typically, a stronger signal can be expected in superficial cell layers than in deeper cell layers (if the protein is

expressed in all cells) explaining the stronger signal in surface cells. However, we were investigating the signal at the tissue interface which is approximately at the same depth. The tissue specific *gsc*:GFP-CAAX expression and the phalloidin staining clearly identifies the tissue interface in our images.

The dotted lines we are using to demark the interface has usually a small offset, otherwise it would mask the signal at the interface. We regret that this was not clear and added this now to the figure captions.

2.) Lifeact.

Convincing images of protrusions: The authors now provided ok images of the protrusions in Fig. 2i

Protrusion dynamics: The authors ignored the request to provide a movie that shows the protrusion dynamics and is the basis for their quantification in Fig. 2j.

We have now provided a movie (Supplementary Video 3) showing actin dynamics of mesendoderm cells in the front and rear of the collective.

3.) Transplantation.

The authors did not address my comments regarding the transplants. Instead of copying my precious comments again, the authors might want to show Fig. 3n to a third person, ask them whether they can see cells with a nuclear label or a membrane label. I cannot. Alternatively, the editors might want to have a look at Fig. 3n and see if they can make out cells with a nuclear label or a membrane label. The concerns are also valid for the *dnRac* transplants.

As previously stated, the main purpose of Fig. 3n was to indicate colour coded tracks related to the graph in Fig. 3m showing internalisation behaviour of neuroectoderm cells. We previously agreed that host and donor cells (all grey LUT) are difficult to distinguish in this case as pointed out by the reviewer and have therefore already provided single colour and merged channels at the last revision to better distinguish the different labels in Suppl. Fig. 4n. We are not sure whether the reviewer looked at those images provided.

In all these experiments, a second marker (or even third marker) was used and is clearly visualised. This labelling technique is routinely used in transplantation assays in zebrafish labs and sufficient to distinguish two cell populations as previously reported (for example, <https://doi.org/10.1016/j.devcel.2022.05.001>;

<https://www.nature.com/articles/ncb3492#Sec34>;

<https://elifesciences.org/articles/42093#s4>;

<https://www.pnas.org/doi/full/10.1073/pnas.1205870109>).

*To make it easier for the readers to identify the strategy for the *cdh1* transplantations, we have now moved some panels from the Supplementary Fig into the main Fig. 3 (m). All single and merged channels are provided in Supplementary Fig. 4n clearly indicating the different labels. We also moved previous panel Fig. 3n into the Supplementary Fig 4 (o) to avoid confusion.*

*Moreover, we have also now performed new *dnRAC* transplantation experiments where we used a third colour (H2A-BFP; cyan) to additionally mark the transplanted *dnRAC* cells as previously requested by the reviewer. We have replaced the panels in Fig. 5i with the new images and included the single and merged channels in Suppl Fig 5h. Importantly, analysing neuroectoderm internalisation in these experiments were*

equivalent to previous quantifications (Figure 1 below), indicating that we can clearly identify the correct cells without additional labelling. Noteworthy, the type of label for transplanted donor cells (membrane, nucleus, etc) is usually not decisive for us, as we use them to qualitatively check the positioning of mesendoderm cells within the embryo and we usually do not use them for analysis. In summary, including additional labels are useful, but optional and not necessary for our analysis and do not change any of our previous conclusions.

NE internalisation depth in previously single labelled ME transplanted cells (and double labelled host) (red, n=3) vs double labelled transplanted cells (and double labelled host) (green, n=1 embryo).

REVIEWERS' COMMENTS

Reviewer #1 (Remarks to the Author):

1. Cdh1

I can confirm the author's original response to this point – it is currently very challenging to import zebrafish lines from abroad due to import restrictions and high cost. If I had been in the same position, I also would have been hesitant to take this route, since even after transport, it would take 4-6 months to grow the line to breeding stage, significantly delaying publication. The transport of adult fish is not feasible with the current regulations. Also, the Cdh1 knock in lines are not specific to the mesendoderm, which would make visualisation of this tissue problematic (other overlying tissues would also be labelled). While there may be some merit in reviewer 2's concerns regarding reporting promoter levels rather than endogenous levels of Cdh1, fusion transgenes like this are regularly used to report the location of proteins within specific populations of cells. In my view, the authors have included sufficient additional experiments to confirm their findings from their fusion transgenic line: Immunohistochemistry staining clearly indicates endogenous levels of protein. Since the authors are interested in relative levels of Cdh1 at the mesendoderm/neuroectoderm interface between the rear and front cells and quantify fluorescence levels within individual embryos, this is an appropriate analysis.

2. Lifeact

The authors supplied the video that reviewer 2 requested

3. Transplantation

Whilst the multiple colours within both host and donor cells make it slightly challenging to understand this data initially, the cell populations are clearly and distinctly labelled in supplementary figure 4n and supplementary figure 5h. This data is appropriate.

In my view, the authors have now addressed reviewer 2's comments and the manuscript should be published.

Clare Buckley

We thank Reviewer 1 for their positive evaluation of our revised experimental work and endorsement of publication.

Reviewer #2 (Remarks to the Author):

Comments for Inman et al.

1. Cdh1 analysis

As stated in the last round of comments, the Cdh1 AB staining is ok. The Cdh1 membrane signal in the revised figures 2 and S3 is still not very convincing but ok. One worry that the authors may want to think about is that the high Cdh1 signal in the NE and the low Cdh1 signal in the ME is mirrored by the high nuclear signal in the NE and

the low nuclear signal in the ME, suggesting that they might be looking at a penetration problem rather than a difference in Cdh1 expression levels – deeper tissues might simply be less well stained.

As we have stated in our previous response, IF staining is usually better close to the surface of the embryo than deeper due to antibody penetration. However, the interface is located at about the same depth (front even closer to surface than rear) where we consistently, over many different batches of embryos, observe differential localisation of cdh1.

2. Lifeact analysis

The movies shown in video 3 are of low quality. It is impossible to make out individual cell protrusions because the temporal resolution is very low and the signal-to-noise is also not great – pseudo-coloring the movies also makes it harder to judge. Nature Methods recently had a comment on how to present fluorescence images, including the advice to always show the grey scale image. Maybe the authors want to heed that advice. The fact that the cells enrich F-actin and form protrusions is supported by the movies so this is ok. However, the authors should label the movies to make it easier for the reader (state the units of the time stamp, provide a scale bar, label the left and right movie in the video itself and not only in the legend).

As suggested by the reviewer, we have provided the time stamp unit, scale bar and labelled front and rear of the collective in movie 3. Converting the colour to greyscale did not improve the labelling of the cells for us, we find that actin-rich protrusions are better highlighted using a graded minimum to maximum colour range.

3. Chimeric analysis

a. Fig. S4n

The cytoplasmic signal for gsc:GFP (host) and the membrane signal for gsc:GFP-caax (donor) cannot be distinguished. There are no cells with GFP on the membrane. GFP looks cytoplasmic, with higher levels in some cells and lower levels in other cells. The membrane label of mRFP (donor) is detectable but the nuclear H2A-RFP label of the host cells is not. The nuclear H2A-BFP label of the donor cells is also not detectable, at best there is some slight enrichment of BFP in some of the cells but there is also the same faint (though a bit weaker) signal in all of the cells. As stated twice before, this is insufficient to support the claims by the authors.

The donor cells are derived from mesendoderm induced embryos which express very high levels of GFP-CAAX signal which explains the non-membrane bound GFP signal. As mentioned before, we have two additional distinct labels to clearly identify donor cells (mRFP and H2A-BFP), so that even in the case where one label might be weak, we can use the other one to clearly identify target cells. As shown in our previous response, single and double label produce similar results. The H2A-mCherry signal in our movies is sufficient to perform single cell tracking of host cells. As stated previously, these labelling techniques are routinely used in many other zebrafish labs, and we are very confident in our results.

b. Fig. 5i and S5h

The labeling of the donor cells with H2A-BFP is acceptable in these panels. It looks like not all donor cells are labeled with H2A-BFP but a large fraction. Also, the membrane GFP labeling of the host cells is visible. So, this is ok and supports the authors conclusion. It is unclear whether all the data were generated using such labeling or not though.

We can assure the reviewer that only data with clear labelling (seen by us and our collaborators) have been used to derive our conclusions.

c. labeling of donor cells

A “qualitative” labeling of donor cells is not sufficient for mosaic analysis nor is it “optional” as the authors propose. Any worm, fly, mouse or fish geneticist would strongly object. One needs to know the genotype of the cells analyzed to correlate genotype with phenotype. It is a bit surprising that this needs to be discussed.

The “qualitative” statement was referring to the type of nuclei or membrane label used to identify the position of cells, not to the cell fate. All transplanted cells are positive for *gsc*:GFP expression, however the strength of the reporter signal per cell is random (as also seen in wildtype embryos).

In summary, the conclusions pertaining to the dnRac experiments are now supported by data. However, the conclusion about the tissue/cell-specific role of *cdh1* is not supported by the data presented and should either be improved or – as suggested before – removed from the paper if the editors agree.

We have provided multiple routes of evidence using several different strategies to determine that *cdh1* plays of crucial role in mesendoderm-neuroectoderm tissue interaction. We are therefore confident that the provided data supports our conclusions.